# Linking Emergent and Natural Languages via Corpus Transfer

**Shunyu Yao**
Princeton University

**Mo Yu**
Wechat AI

**Yang Zhang**
MIT-IBM Watson AI Lab

**Karthik Narasimhan**
Princeton University

**Joshua B. Tenenbaum**
MIT

**Chuang Gan**
MIT-IBM Watson AI Lab

## Abstract

The study of language emergence aims to understand how human languages are shaped by perceptual grounding and communicative intent. Computational approaches to emergent communication (EC) predominantly consider referential games in limited domains and analyze the learned protocol within the game framework. As a result, it remains unclear how the emergent languages[1] from these settings connect to natural languages or provide benefits in real-world language processing tasks, where statistical models trained on large text corpora dominate. In this work, we propose a novel way to establish such a link by *corpus transfer*, i.e. pretraining on a corpus of emergent language for downstream natural language tasks, which is in contrast to prior work that directly transfers speaker and listener parameters. Our approach showcases non-trivial transfer benefits for two different tasks – language modeling and image captioning. For example, in a low-resource setup (modeling 2 million natural language tokens), pre-training on an emergent language corpus with just 2 million tokens reduces model perplexity by 24.6% on average across ten natural languages. We also introduce a novel metric to predict the transferability of an emergent language by translating emergent messages to natural language captions grounded on the same images. We find that our translation-based metric highly correlates with the downstream performance on modeling natural languages (for instance $\rho = 0.83$ on Hebrew), while topographic similarity, a popular metric in previous work, shows surprisingly low correlation ($\rho = 0.003$), hinting that simple properties like attribute disentanglement from synthetic domains might not capture the full complexities of natural language. Our findings also indicate potential benefits of moving language emergence forward with natural language resources and models[2].

## 1 Introduction

The recent remarkable progress in NLP (Devlin et al., 2018; Radford et al., 2019; Brown et al., 2020) benefits from huge text corpora, on which powerful models learn rich statistical patterns *tabula rasa* and adapt to downstream tasks. But in stark contrast, human language acquisition (De Villiers et al., 1978) does not start with passive and complex corpora like Wikipedia. Instead, children learn language through a cumulative series of interactions with other people (Bruner, 1985; Barton, 1994) grounded in the physical and social world (Harnad, 1990; Vogt, 2002; Alomari et al., 2017). Incorporating such cognitive insights with recent machine learning advances holds great promises towards more functional (Wittgenstein, 1958; Wagner et al., 2003) and generalizable (Lake et al., 2017) language agents.

The study of emergent communication (EC) (Cangelosi & Parisi, 2002; Kirby, 2002; Wagner et al., 2003; Lazaridou & Baroni, 2020) is one promising direction toward this motivation, where a commu-

---

[1]In this work "emergent language" means generated messages from an EC game speaker. The term also refers to children's language development in psychology (Bohlmann & Downer, 2016; Zajicek-Farber, 2010).

[2]Code at https://github.com/ysymyth/ec-nl and correspondence to shunyuy@princeton.edu. Work partly done during the first author's remote internship at MIT-IBM Watson AI Lab.

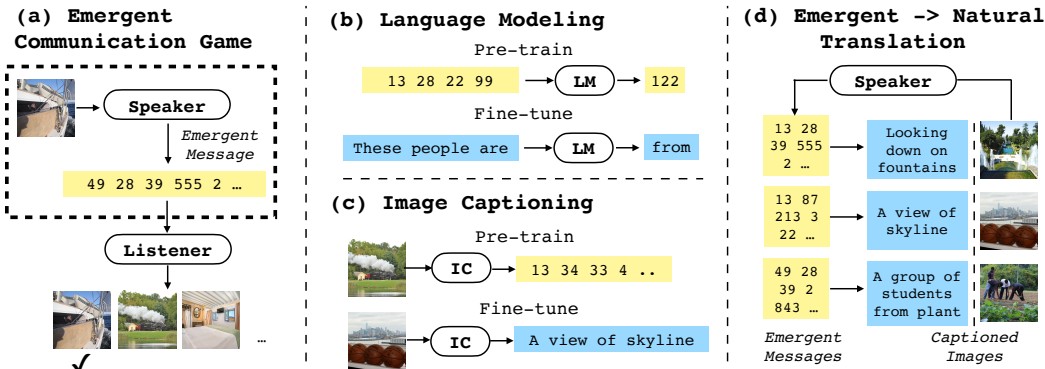

Figure 1: Our framework. (a) We train a referential game on natural images, and **use the trained speaker to generate a corpus of emergent messages** (bold box) to pre-train for (b) language modeling and (c) image captioning. We also propose to (d) translate emergent language (yellow) into corresponding natural language (blue) with same perceptual grounding to evaluate their closeness.

nication protocol is shaped through multi-agent interactions with perceptual grounding. A typical setup is the image referential game (Figure 1(a)), where a speaker generates a discrete sequence of tokens based on an input image, a listener is challenged to select the input out of distractors based on the message, and both networks are optimized jointly via game success signals. By studying these games, researchers are interested in the emergence of desirable properties resembling natural language, such as game success generalization (Kharitonov & Baroni, 2020; Lazaridou & Baroni, 2020) and compositionality (Smith et al., 2003; Andreas & Klein, 2017; Kirby et al., 2015; Lazaridou et al., 2018; Li et al., 2020b). However, these properties are mostly defined and analyzed within each individual game framework. For example, generalization is usually measured as game success with novel inputs, but it remains unknown how the emergent coding can be useful beyond the particular game (e.g. useful for natural language tasks). Moreover, compositionality is often approximated by attribute-wise disentanglement in simple environments, which could be drastically different from the structural properties of natural language (Dankers et al., 2021). Such gaps pose difficulty in objectively understanding the emergent properties with respect to the full complexities of natural language, and assessing the universality of various claims.

In this work, we aim to address these issues by linking emergent languages with natural languages. Concretely, we investigate if modeling an emergent language provides transferable benefits for downstream natural language tasks — language modeling (Figure 1(b)), image captioning (Figure 1(c)) — and if downstream performances in turn allow us to understand and analyze properties of emergent language beyond the EC game and toward natural language. The key technique is **corpus transfer**, i.e. by pre-training a model on a corpus of emergent messages produced by a trained emergent speaker, and fine-tuning the model on downstream tasks with natural language data. This approach integrates inspirations from prior work like Papadimitriou & Jurafsky (2020), which proposes the transfer scheme from synthetic to natural corpora, and Li et al. (2020b), which aims to improve few-shot translation by transferring the trained EC speaker and listener model weights. Through a series of experiments, we find that corpus transfer is helpful when the downstream natural language resource is limited. For example, in a low-resource setup of modeling two million natural language tokens, such a transfer scheme reduces the test perplexity by $24.6\%$ on average versus training from scratch, across ten different downstream languages. We also establish the non-triviality of such a transfer performance by comparing to other synthetic and natural source corpora, as well as multiple ablation studies on the EC and downstream transfer setups to help understand the transferability of emergent language. Notably, our method of corpus transfer significantly outperforms directly transferring the trained emergent speaker model (Li et al., 2020b), demonstrating that modeling the emergent language could yield greater usefulness than directly transferring the EC agents.

While our transfer experiments demonstrate that emergent languages can be leveraged and understood outside the game and towards real-world tasks, the pre-training and fine-tuning pipeline is computationally expensive as a scalable evaluation scheme for a population of emergent languages. Instead, we are interested in developing a simpler metric to predict the transferability of emergent language, which pertains to properties of natural language and may challenge metrics defined within

the game framework (e.g. game accuracy). Therefore, we propose a novel metric for emergent languages based on how easily it can be translated to the corresponding natural language grounded on the same perceptual input (Figure 1(d)). Concretely, we employ the trained speaker to produce emergent messages based on a small set of unseen images with natural language captions, and train a translation model on paired emergent and natural captions. We show that the translation score (such as ROUGE_L (Lin, 2004)) can better predict (e.g. Pearson correlation $\rho = 0.83$ for modeling Hebrew) the transfer benefits of different emergent languages than two commonly adopted metrics in prior literature – game success with novel inputs ($\rho = 0.74$), and topographic similarity (Brighton & Kirby, 2006; Lazaridou et al., 2018) ($\rho = 0.003$). These results call for a re-thinking of what properties we should evaluate for an emergent language corpus, and how we should evaluate them. For example, game accuracy may be confounded by the listener performance, and topographic similarity, a metric of attribute disentanglement, can be flawed for measuring the compositional structure of real language (Kirby, 2001; Goldberg, 2015; Steinert-Threlkeld, 2020).

In summary, our work takes a step toward bridging the gap between emergent and natural languages. We believe that shifting from synthetic setups and in-game evaluations to leveraging downstream natural language tasks and state-of-the-art language models could potentially lead to more general insights and greater practical impact for EC and NLP research.

## 2  RELATED WORK

**Emergent Communication and Evaluation**   A large body of prior work qualitatively analyzes the evolved message structures when the EC games are set up with synthetic environments (Mordatch & Abbeel, 2018; Lazaridou et al., 2018) or even with real images (Havrylov & Titov, 2017). Alternative evaluations and examinations include ease of teaching (Li & Bowling, 2019) and symbol usage purity (Lazaridou et al., 2016). While these schemes align the emergent language with game success or different aspects of game input, our work evaluates the emergent language outside the game framework, instead via the corpus transfer performance to natural language tasks and the translation performance to natural language.

**Linking Synthetic and Natural Language**   Papadimitriou & Jurafsky (2020) find that recurrent networks pretrained on non-linguistic corpora (e.g. music, code, regular languages) facilitate natural language modeling. In an opposite direction, Lu et al. (2021) observe that language pre-training improves Transformer's performance with several non-language tasks (logical, vision, protein) even when most weights are frozen, indicating that learning language structure could yield generally useful computational subroutines. Andreas et al. (2017) propose to translate a differentiable communication channel in deep multi-agent policies to natural language by collecting human-human communication in the same games. However, their approach focused on *interpreting* a *continuous* channel, while we focus on *evaluating* emergent messages with *discrete* structures.

**Use of Emergent Communication beyond games**   There have been efforts to incorporate communication signals and natural language supervision to better communicate (Lowe et al., 2020), avoid language drift (Lazaridou et al., 2020), or to improve NLP tasks like vision-language navigation (Fried et al., 2018), translation (Lee et al., 2018), and image captioning (Havrylov & Titov, 2017). Different from these setups where the speaker is also grounded on natural language annotations, Li et al. (2020b) propose to separate the EC game, where no natural language is involved, and fine-tuning on a downstream translation task with limited natural language data. Despite being the closest work to ours, such a *model transfer* approach is fundamentally different from our *corpus transfer* in several ways. First, while Li et al. (2020b) mainly aim to improve downstream tasks, we envision transfer as a novel means to evaluating and analyzing emergent languages. Also, the emergent *language* is a representation with properties beyond the parameters of specific EC models (Section 3.1). Last but not least, corpus transfer has several practical advantages over model transfer (Section 4.3).

## 3  BACKGROUND

### 3.1  EMERGENT COMMUNICATION (EC) GAME

As shown in Figure 1(a), we consider a typical speaker-listener referential game (Lazaridou et al., 2016; 2018; Li & Bowling, 2019) on a set of $N$ image features $\mathcal{D}_I = \{\mathbf{I}_1, \cdots, \mathbf{I}_N\}$. At each training

step, the speaker takes an input image feature $\mathbf{I}_i$ and generates a discrete message $\mathbf{M}_i \in [V]^T$ using the Gumbel-Softmax trick (Jang et al., 2016), where $V$ is the vocabulary size and $T$ is the sequence length limit. For simplicity denote $\mathbf{m} = \mathbf{M}_i$ so that $m_t = M_{i,t}$ denotes the $t$-th token of the message $\mathbf{M}_i$. Then

$$\mathbf{hs}_0 = I_i, \quad \mathbf{hs}_t = \mathrm{GRU}_{spk}\left(m_{t-1}, \mathbf{hs}_{t-1}\right) \ (t > 0),$$
$$m_0 = [\mathrm{CLS}], \quad m_t = \mathrm{Gumbel\text{-}Softmax}\left(\mathrm{MLP}_{spk}(\mathbf{hs}_t)\right) \ (t > 0). \tag{1}$$

Here $\mathbf{hs}_t$ denotes speaker hidden states. Next, the listener takes the message $\mathbf{m}$, and tries to guess the right image $\mathbf{I}_i$ out of a set of $K$ confounding images $C_i = \{\mathbf{I}_{j_1}, \cdots, \mathbf{I}_{j_K}\} \subset \mathcal{D}_I - \{\mathbf{I}_i\}$. To do so, it uses another GRU layer to turn the message $\mathbf{m}$ into a hidden vector $\mathbf{hl}_T$

$$\mathbf{hl}_0 = \mathbf{0}, \quad \mathbf{hl}_t = \mathrm{GRU}_{lsn}\left(m_t, \mathbf{hl}_{t-1}\right) \ (t > 0). \tag{2}$$

Based on $\mathbf{hl}_T$, the listener assigns a score for each candidate image based on inverse square error (Lee et al., 2018), then selects the image by Softmax sampling across the scores. The speaker and listener are jointly optimized by minimizing the cross-entropy loss of image selection:

$$\mathrm{score}(\mathbf{I}) = ||\mathbf{hl}_T - MLP_{lsn}(\mathbf{I})||_2^{-2}, \tag{3}$$
$$p(\mathrm{guess} = \mathbf{I}) = \mathrm{softmax}(\mathrm{score}(\mathbf{I})) \quad (\mathbf{I} \in \{\mathbf{I}_i\} \cup C_i), \tag{4}$$
$$\mathcal{L}_{EC} = -\mathbb{E}_{\mathbf{I}_i, C_i} \mathbb{E}_{\mathbf{M}_i} \log p(\mathrm{guess} = \mathbf{I}_i). \tag{5}$$

After the game is trained, the speaker can be employed to sample a corpus of *emergent language* $\mathcal{D}_M = \{\mathbf{M}_1, \cdots, \mathbf{M}_N\}$ based on input images. Though the emergent language is generated by a particular EC speaker, the representation could be potentially learned and used by new agents, with more general properties beyond the parameters of a particular speaker architecture. Thus, our work focuses on evaluating and analyzing the properties of the emergent language (e.g. corpus transfer) rather than the emergent communication models (e.g. model transfer).

## 3.2 Evaluating Emergent Language

A central question in emergent communication is how to evaluate the quality of the emergent corpus $\mathcal{D}_M$. Two commonly used metrics in prior work are game accuracy and topographic similarity.

**Game Accuracy with Novel Objects** The most straightforward metric is the accuracy of playing the referential game $\mathbb{E}_{\mathbf{I}_i, C_i, \mathbf{M}_i}\left[\mathbf{1}_{\mathrm{guess} = \mathbf{I}_i}\right]$. Prior work (Cogswell et al., 2019; Li & Bowling, 2019) usually considers how well the speaker and listener generalize to novel inputs, and finds that such a generalization ability might not correlate with other properties such as compositionality (Chaabouni et al., 2020; Kharitonov & Baroni, 2020). Note that this metric involves both the speaker and listener, while the emergent language corpus $\mathcal{D}_M$ is generated solely by the speaker.

**Topographic Similarity** The topographic similarity (Brighton & Kirby, 2006; Lazaridou et al., 2018) measures how messages align with the meaning representations. More concretely, let $CS_{ij} = -\mathbf{I}_i \cdot \mathbf{I}_j / (||\mathbf{I}_i||_2 ||\mathbf{I}_j||_2)$ be the negative cosine similarity of image features $\mathbf{I}_i$ and $\mathbf{I}_j$, and $ED_{ij}$ be the Levenshtein distance (Levenshtein et al., 1966) between messages $\mathbf{M}_i$ and $\mathbf{M}_j$. The metric is defined as the Spearman rank correlation of the two distance metrics:

$$r_{ED,CS} = \rho_{R(ED),R(CS)} = \frac{\mathbb{E}_{i,j}\left[(R(ED_{ij}) - \mu_{R(ED)})(R(CS_{ij}) - \mu_{R(CS)})\right]}{\sigma_{R(ED)}\sigma_{R(CS)}} \tag{6}$$

Here $R(\cdot)$ denotes the rank of the element in the list. This metric intuitively measures *disentanglement*: different attributes of the input should be expressed in different token positions, and each token only describes one attribute (e.g. "big red cube" and "big blue cube"). Such a property has been claimed (Lazaridou et al., 2018; Li & Bowling, 2019) as connected to *compositionality*, a key structural property of natural language. However, this metric is too rigid in its definition of compositionality, ignoring aspects like argument structure, context or morphology which play a key role in determining the combination of word semantics (Goldberg, 2015).

We note that experiments around both these metrics are within the *game framework* – accuracy is based on *game success*, while topographic similarity measures how emergent messages align with *game input representations*. This leaves two questions unsolved: (i) can emergent languages be used outside the game? (ii) if so, would these metrics predict the usefulness of emergent languages for downstream tasks? We tackle these two questions in Sections 4 and 5 respectively.

## 4 EMERGENT CORPUS TRANSFER FOR NATURAL LANGUAGE TASKS

In this section, we present initial evidence that a *corpus* of emergent language $\mathcal{D}_M$ can benefit two types of downstream tasks involving natural language: language-only task (language modeling, Section 4.1) and vision-language task (image captioning, Section 4.2). Moreover, we show such a benefit is most significant when the natural language resources are limited, which indicates that emergent language is potentially valuable for low-resource languages and tasks. We then analyze what contributes to such a transfer via ablation studies in Section 4.3.

### 4.1 LANGUAGE MODELING

Inspired by Papadimitriou & Jurafsky (2020), we consider the task of natural language modeling, and explore if pretraining a language model on a corpus of emergent messages can reduce the downstream perplexity in a low-resource setup (Figure 1(b)), and how emergent language compares to other sources of synthetic and even natural language corpora.

**Pre-training Corpora**   We compare three source corpora: (i) a Spanish Wikipedia corpus (**es**) culled to 50,000 vocabulary size using the pre-processing script from Papadimitriou & Jurafsky (2020), (ii) a regular language of well-balanced brackets (**paren-zipf**) with the Zipf unigram distribution as **es**, and (iii) an emergent language corpus (**ec**). **paren-zipf** is the best-performing source corpus from Papadimitriou & Jurafsky (2020) not created by humans (e.g. natural language, music, or code) with the inductive bias of *hierarchical structure*, and **es** should set an upper bound of the transfer performance for both **paren-zipf** and **ec**. We vary the source corpus size from 2, 5, 10, 15 to 30 million tokens to understand how the transfer results change with respect to the pre-training scale.

**Fine-tuning Corpora**   We scrape Wikipedia corpora of 10 languages to test downstream transfer: Danish (**da**, IE-Germanic), Basque (**eu**, Basque), Japanese (**ja**, Japanese), Romanian (**ro**, IE-Romance), Finnish (**fi**, Uralic), Indonesian (**id**, Austronesian), Kazakh (**kk**, Turkic), Hebrew (**he**, Afro-Asiatic), Urdu (**ur**, IE-Indic), and Persian (**fa**, IE-Iranian). They come from diverse linguistic families with different levels of resource richness. Each corpus has 2 million tokens and a vocabulary size of 50,000, and is pre-processed (to remove noise) via the same script to process **es**.

**Implementation**   We train the EC game and generate **ec** based on the Conceptual Captions dataset (Sharma et al., 2018), using more than 2.8 million natural images in the wild. Note that we do not use the natural language captions in the dataset for EC game training. We take the 512-dim pre-trained ResNet-18 (He et al., 2016) features before the classification head as input image features $\mathbf{I}_i$. Other architecture and training details mainly follow Li et al. (2020b), and by default $V = 4035$, $T = 15$, $K = 256$. For language modeling, we adopt a Transformer (Vaswani et al., 2017) with 6 decoder layers and 6 attention heads, and pre-train on each source corpus for 3,000 steps with batch size 32, input length 1,000, and learning rate $5 \times 10^{-4}$. For fine-tuning and training from scratch on downstream corpora, the batch size is $8$ and learning rate is $10^{-4}$. Hyperparameters are chosen after grid search. We report the test perplexity at the best validation loss.

**Results**   As can be seen from Figure 2, pre-training on **ec** (blue) significantly reduces the perplexity compared with training from scratch (red, dotted line with constant value), with an average reduction of $24.6\%$ with merely 2 million pre-training tokens. This is a positive signal that **a corpus of emergent language could be useful beyond the communication game itself**.

Comparing two synthetic pre-training corpora, **ec** performs better than or comparable to **paren-zipf** throughout different source sizes and downstream languages, with the exception of Basque and Finnish when pre-training size is more than 15 million. Given the strong inductive bias in **paren-zipf** (hierarchical structure and unigram information from real language), and the fact that **ec** is generated by just a single-layer GRU speaker without such explicit bias, such a result hints that **ec** may possess non-trivial structural properties due to the communication signals and perceptual grounding.

On the other hand, the natural language corpus **es** usually achieve a lower perplexity than both **ec** and **paren-zipf** as may be expected. Most surprisingly, when pre-training only on 2M tokens, **ec** consistently beats **es** in terms of transfer performance, and their performances are comparable on most languages using 5 million source tokens. To offer one plausible explanation, **ec** could contain

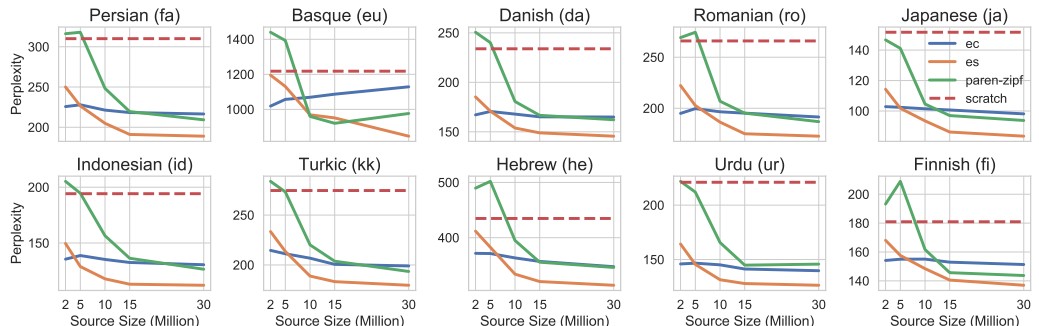

Figure 2: Test perplexity for language modeling on ten natural languages, when either pre-trained on a corpus of tokens from EC (**ec**, blue), Spanish (**es**, orange), well-balanced brackets (**paren-zipf**, green) or without any pre-training (**scratch**, red, dotted, constant with respect to source sizes).

| **Metrics** | coco (5k) | | | coco (50k) | | | coco (full) | | |
|---|---|---|---|---|---|---|---|---|---|
| | **base** | **+ec** | **+nl** | **base** | **+ec** | **+nl** | **base** | **+ec** | **+nl** |
| BLEU4 | 14.1 | 15.3 | 18.5 | 26.3 | 27.2 | 28.2 | 35.8 | 36.2 | 36.0 |
| ROUGE_L | 39.8 | 41.0 | 44.4 | 50.0 | 50.7 | 51.5 | 56.9 | 57.0 | 57.1 |
| CIDEr | 30.8 | 35.7 | 48.0 | 78.3 | 82.8 | 88.8 | 117.5 | 118.1 | 118.5 |

Table 1: Results on image captioning transfer using three fine-tuning dataset sizes. **base**, **+ec**, **+nl** denote training from scratch and pre-training using EC captions and NL captions, respectively.

properties closer to everyday conversations – smaller vocabulary, simpler structures (generated by one-layer GRU) and repetitive (all about image references) – easier for language modeling, compared to the complicated structures and statistical patterns of large text corpora like Wikipedia **es**. However, the relative simplicity of **ec** also prevents the transfer from scaling as well as **es** - comparing pretraining on 30 versus 2 million tokens, **ec** only reduces perplexity by $2.4\%$ on average, while **es** reduces $23.8\%$. Such a mixed finding points to potential in combining the strengths of simpler emergent corpora (easier to acquire and benefit from) and noisy natural language corpora (more structures and complexities) for more efficient language learning. Such an idea of mixing NLP and EC training has been useful for some NLP (Fried et al., 2018; Lee et al., 2018) and EC (Lowe et al., 2020; Lazaridou et al., 2020) tasks. Another important direction would be on how to evolve emergent languages with more complexities that resemble and transfer to natural language, where existing approaches to improving and regularizing EC (Mu & Goodman, 2021; Guo et al., 2019; Li & Bowling, 2019; Dessì et al., 2021; Luna et al., 2020) could be tested through our transfer scheme.

## 4.2 IMAGE CAPTIONING

The language modeling transfer experiment leverages the *structural* properties of the source corpus, but has little to do with the *semantics* of the source corpus. For example, a regular language like **paren-zipf** has no real meanings, but still reduces downstream perplexity due to its hierarchical structure. Therefore, we also consider transfer to image captioning, a classical vision-language task where a model has to ground meaning of the words to the images. As shown in Figure 1(c), we pre-train an image captioning model on unlabeled images to generate EC captions, then fine-tune on downstream annotated images to generate natural language captions. We note that such a transfer scheme cannot work for other synthetic corpora such as music, code, or **paren-zipf** as they are not grounded on perceptual stimuli.

**Pre-training Data** We use the EC speaker trained on Conceptual Captions to generate an EC caption $\mathbf{M}_i$ based on each ResNet-18 image feature $\mathbf{I}_i$ in the dataset. As image captioning requires fine-grained information beyond a global feature, we extract detection features $\mathbf{I}'_i$ for each image via pre-trained Faster R-CNN (Ren et al., 2015) and ResNet-101 (He et al., 2016), and pre-train an image captioning model to generate EC captions based on detection features ($\mathbf{I}'_i \mapsto \mathbf{M}_i$). To set up an upper

| Method | LM (ro) | LM (he) |
|--------|---------|---------|
| EC pretrain | **198 (4)** | **375 (14)** |
| from scratch | 265 (1) | 446 (8) |
| random speaker | 313 (21) | 575 (24) |
| random input | 245 (28) | 489 (77) |
| permuted EC | 211 (1) | 383 (4) |
| model transfer (GRU) | 655 (40) | 1105 (6) |
| corpus transfer (GRU) | 553 (53) | 1056 (55) |

Table 2: Ablations on EC game, corpus, and transfer setups. Standard deviations in parentheses. The language model is an GRU for last two rows and a Transformer for others.

| Metrics | LM (ro) | LM (he) |
|---------|---------|---------|
| Accuracy | 0.672 | 0.737 |
| Topographic | 0.030 | 0.003 |
| Translation | **0.757** | **0.829** |

Table 3: Pearson correlations between different metrics (validation accuracy, topographic similarity, emergent to natural language translation ROUGE_L) and downstream language modeling performance (negated perplexity).

bound for transfer performance, we also use the English captions $\mathbf{NL}_i$ in the Conceptual Captions dataset for pre-training ($\mathbf{I}'_i \mapsto \mathbf{NL}_i$).

**Fine-tuning Data**   We use the MS-COCO dataset (Lin et al., 2014) for fine-tuning, which consists of around 500,000 pairs of images and English captions. We use the full training set, or a subset with 5,000 or 50,000 samples to study the transfer benefit when natural language annotation is limited.

**Implementation**   We use a Transformer with 3 encoder layers and 6 decoder layers as the image captioning model, and perform pre-training and fine-tuning in a seq2seq framework (Ott et al., 2019). Notably, we only transfer the encoder weights, and fine-tune the whole network end-to-end with a learning rate of $3 \times 10^{-4}$, as we notice transferring the full weights leads to worse performance. We report BLEU4 (Papineni et al., 2002), ROUGE_L (Lin, 2004), and CIDEr (Vedantam et al., 2015) scores for the test split based on best validation loss.

**Results**   As shown in Table 1, when fine-tuning on 5,000 or 50,000 samples, pre-training on either EC or NL captions both significantly improve the image captioning performance. For example, with 50,000 samples, EC pre-training can improve the BLEU-4 score by +0.9 points, while NL pre-training further improves +1.0 points. The fact that EC pre-training can provide half the benefit of NL pre-training is surprising, because the former requires no language annotation - both training the EC games and generating the EC captions can be done on unlabeled images in the wild, while the NL pretraining leverages more than 2.8 million English sentences.

However, when the full MS-COCO dataset is used, even pretraining on 2.8 million English captions is not significantly helpful, possibly because Conceptual Captions collect noisy Internet sentences while MS-COCO annotates captions with less diversity (Wang et al., 2020) so a model trained from scratch on the full dataset can capture most of the output patterns. Other vision-language tasks where natural language annotations are harder or more limited (e.g. instruction following, navigation) would potentially benefit more from EC pre-training, where we can turn abundant unlabeled visual stimuli into annotations in the emergent language.

## 4.3   WHAT CONTRIBUTES TO SUCCESSFUL CORPUS TRANSFER?

We have shown that emergent languages can provide transfer benefit for natural language tasks, but it is still unknown what properties of emergent languages contribute to such a benefit. Hence we ablate several key aspects of the EC game training and EC corpus generation, and examine the effect via transfer (with 15 million pre-training tokens) to two example languages, Romanian (**ro**) and Hebrew (**he**). We leave ablation studies on the image captioning experiment in Section A.3.

**Communication and Perceptual Stimuli in EC Game**   The emergent language stems from two elements in the game that human language acquisition relies on: perceptual grounding (input image features) and communication (referential game). For ablation, we first consider an untrained GRU speaker with randomized parameters (**random speaker**), and use it to sample tokens conditioned on image features according to (1). As Table 2 shows, pre-training on such a corpus leads to even worse

performance than training from scratch, because an untrained speaker has highly random generations with no clear structure.

On the other hand, to ablate the effect of visual stimuli, we sample a set of $N$ random vectors from $\mathcal{N}(\mu(\mathcal{D}_I), \sigma(\mathcal{D}_I))$, and replace it for $\mathcal{D}_I$ as inputs of the EC game (**random input**). We then use the trained speaker to generate a EC corpus based on the same random vectors. It turns out only one out of three game trials develop a game accuracy $> 90\%$, and other two trials cannot learn an accuracy beyond $5\%$. On Hebrew, the "successful" trial achieves a transfer perplexity of 388, which is still worse than EC corpus on visual inputs, while other two trials yield significantly worse perplexities (573, 509). So it turns out communicating visual stimuli instead of random inputs is more robustly successful, and more helpful for transfer even when communication can be successful.

**Sentence Structure in EC Corpus**    To investigate how structured EC sentences are, we independently shuffle each EC sentence in $\mathcal{D}_M$ to form a new-pretraining corpus (**permuted EC**), and find in Table 2 that it indeed leads to a consistent worse performance, though better than training from scratch. This indicates the speaker develops a coding scheme with sentence structures beyond bag-of-words.

**Model vs. Corpus Transfer**    Li et al. (2020b) propose to transfer the EC models for a downstream translation task, whereas we leverage a speaker-generated EC corpus to pre-train with architectures that could be more flexible and powerful, which leads to several significant practical advantages. First, EC model architectures might not apply to many downstream tasks (e.g. image captioning with multiple input features), but corpus transfer could still be readily deployed (Section 4.2). On other tasks (e.g. language modeling) where model transfer could be implemented, its performance is still constrained by the simple EC model architectures[3]. As shown in Table 2, directly fine-tuning the one-layer speaker GRU (**model transfer (GRU)**) indeed leads to much worse perplexities than corpus transfer with a Transformer (**EC pretrain**), but more interestingly, is also worse than corpus transfer with the same GRU architecture (**corpus transfer (GRU)**), possibly due to the benefit of self-distillation (Zhang et al., 2019; Allen-Zhu & Li, 2020). In summary, our corpus transfer approach better utilizes the emergent communication for downstream tasks than model transfer.

# 5    EMERGENT-NATURAL LANGUAGE TRANSLATION AS A METRIC

Section 4.3 presents initial evidence that the quality of an emergent language correlates with the transfer performance, by showing ablated emergent languages transfer worse. However, directly using transfer as an evaluation scheme for emergent languages has certain deficiencies. First, the pre-training and fine-tuning optimizations are computationally expensive. Second, transfer performance across different downstream tasks might induce inconsistency. Hence we propose a cheaper and simpler evaluation for emergent languages by measuring how easily it translates to the corresponding natural language grounded on the same input, and show that it better correlates with the transfer performance than existing metrics across different tasks.

More concretely, we use a small set of captioned images, where image $\mathbf{I}_i$ is associated with a natural language caption $\mathbf{NL}_i$. We also use the trained speaker to generate emergent message $\mathbf{M}_i$ based on each $\mathbf{I}_i$, and train a translation model to map emergent sentences to natural sentences $\mathbf{M}_i \mapsto \mathbf{NL}_i$. Intuitively, a higher translation score means the emergent and natural sentences are closer in structure and semantics, similar to how French-English translation might be easier than Chinese-English.

**Implementation**    To estimate the correlation between metrics and transfer performance requires a population of emergent languages with different speakers.    Thus we choose 5 EC game setups with varying vocabulary and sequence length limits ($(V, T) \in \{(4035, 5), (4035, 15), (4035, 25), (1000, 15), (10000, 25)\}$), and for each setup we run 4 trials for 2,000 steps. We consider checkpoints every 200 steps, summing to $5 \times 4 \times 10 = 200$ checkpoints. Due to computational limits, we train EC games on a small subset of 50,000 MS-COCO images, and use another subset of 50,000 MS-COCO image-caption pairs to calculate the translation metric. For translation, we use the same seq2seq Transformer for image captioning, but only train for 2 epochs, so that an emergent language more similar to the natural language can perform better with limited

---

[3]We note that unlike most NLP tasks, EC model architectures might not arbitrarily scale up due to optimization challenges with discretization. For example, we have tried 2/3-layer GRUs for EC, but the transfer is worse.

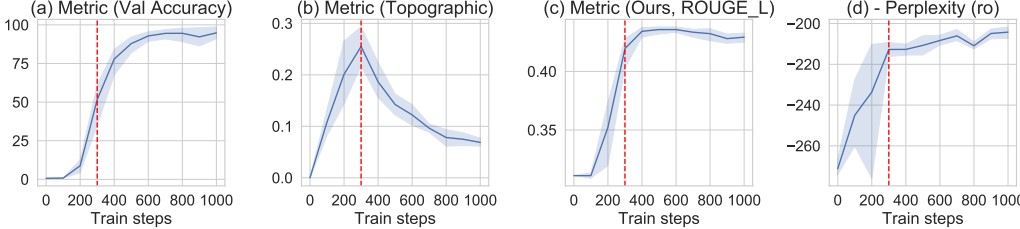

Figure 3: Different metrics and downstream Romanian perplexities (negated) with respect to training steps, averaged over four trials with vocabulary limit 10000 and sequence length limit 15. Our metric better correlates with downstream performance across time steps.

training. We use ROUGE_L (Lin, 2004) as the translation metric, and find other metrics with similar results. For downstream language modeling, we use ImageNet (Deng et al., 2009) to generate a corpus of 15 million tokens and fine-tune on Romanian (**ro**) and Hebrew (**he**).

**Results**   We compare with the two existing metrics, validation accuracy and topographic similarity (see Section 3.2). Table 3 shows how different metrics correlate with downstream performance across all checkpoints form all trials[4]. Surprisingly, we find little correlation between topographic similarity and downstream performance, while our translation metric has a better correlation ($\rho_{ro} = 0.757, \rho_{he} = 0.829$) than validation accuracy ($\rho_{ro} = 0.672, \rho_{he} = 0.737$).

To better understand the results, we take all 4 trials from a specific game setup ($V = 10000, T = 15$) and plot in Figure 3 how the three metrics (accuracy, topographic, translation ROUGE_L) as well as the downstream Romanian performance change with training steps (similar analysis of more setups (vocabulary size, sequence length) in Section B.4). We observe that our metric (Figure 3(c)) aligns with the downstream performance (Figure 3(d)) in that both reach a high level around 300 steps (red dash line) and maintains a similar level afterwards. The validation accuracy ((Figure 3(a))) has a generally similar tendency as these two metrics, with the difference being that it is only around 50% at step 300 and far from convergence, possibly due to the underdevelopment of the listener. A similar observation was made in Li et al. (2020b). Such a dependency on the listener renders game success unreliable for the evaluating emergent languages produced by the speaker alone.

On the other hand, the topographic similarity (Figure 3(b)) peaks as early as step 300, after which the degradation fails to explain the downstream performance. To illustrate the misalignment, consider the "best" language with respect to topographic, i.e. a fully disentangled coding scheme (e.g. "red/blue cube/sphere"). Such a corpus would essentially have a maximum possible entropy, pre-training on which should not help with any downstream tasks. Our finding suggests that **simple measurements of rigid disentanglement fall short of estimating the compositionality of real language**, and we may need to take into account other structural properties of natural language, e.g. stable irregularity (Kirby, 2001), or the role of argument structures, context and morphology (Goldberg, 2015; Steinert-Threlkeld, 2020).

## 6   CONCLUSION

While previous research in language emergence tends to investigate different setups and how they affect certain metrics, such a paradigm cannot provide robust and valuable insights if the metrics are flawed. Thus, our work calls for a *paradigm shift* in how we evaluate and analyze language emergence — by stepping from metrics within each EC framework or over-approximate NL properties, and instead directly linking to NL by transfer or translation. Such a paradigm shift enables several brand new and exciting future directions — how to improve or regularize EC through the lens of our metrics, how to push EC to be more useful for NLP[5], and how to design more fine-grained metrics for specific NL properties. We believe progress in these directions will further develop a *synergy* between EC and NLP research and bring benefits and insights for both sides.

---

[4]For topographic, we exclude 9 checkpoints with the metric undefined due to zero variance of edit distances.
[5]See relevant discussions at the end of Section 4.1 and Section 4.2.

## ACKNOWLEDGEMENTS

This work was supported by MIT-IBM Watson AI Lab and its member company Nexplore, ONR MURI (N00014-13-1-0333), DARPA Machine Common Sense program, ONR (N00014-18-1-2847) and MERL. The information, data, or work presented herein was also funded by the Advanced Research Projects Agency-Energy (ARPA-E), U.S. Department of Energy, under Award Number DE-AR0001210. SY and KN also acknowledge support from the National Science Foundation under Grant No. 2107048. The views and opinions of authors expressed herein do not necessarily state or reflect those of the United States Government or any agency thereof.

## ETHICS STATEMENT

Our work is a preliminary step toward aligning the language of machines and humans, which has potential positive social impacts in several ways. First, shaping machine communications towards the structure of natural language contributes to human and AI alignment in terms of values and understanding of the world. Second, emergent languages could yield transfer benefits for low-resource languages and tasks, contributing to linguistic diversity and inclusion. We could not think of negative ethics impact for now, as the emergent language is still very different from natural language.

## REPRODUCIBILITY STATEMENT

Our research in based on public codebases and datasets, with implementation details described throughout the paper. We have conducted reasonable hyperparameter searches and ablation studies to make sure experiment conclusions are fair. In the Appendix, we further provide links to all used code and data resources, and additional details including hyperparameter search setups and resource requirements for each experiment. We will clean and release the code upon acceptance.

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

# A    IMPLEMENTATION DETAILS

## A.1    EMERGENT COMMUNICATION GAME

We adapt the public code[6] from Li et al. (2020b) and mostly follow their default setups. For training, we use a batch size of 256, and each batch element contains one input images and other 255 distractor images. Since the Conceptual Captions dataset has more than 2.8 million images, random sampling a batch of data is computationally costly. So for each batch, we first sample 50,000 images from the whole dataset, then sample input and distractor pairs from this subset. We use an Adam optimizer with learning rate $10^{-3}$. We use a soft version of Gumbel-softmax with temperature 1, and have tried hard Gumbel-softmax and found it not further helpful for downstream performance. Each game training only takes less than 12 hours using one GeForce RTX 2080 GPU.

## A.2    LANGUAGE MODELING

We use the public script[7] from Papadimitriou & Jurafsky (2020) to pre-process Wikipedia corpora of different languages, using the default setup of culling to 50,000 vocabulary size. We hand-pick downstream languages to make sure they represent different linguistic families.

We use the language modeling script[8] from Huggingface (Wolf et al., 2019) for both pre-training and fine-tuning.

We have tried grid search for the pre-training learning rate $(10^{-3}, 5 \times 10^{-4}, 10^{-4})$ and batch size $(4, 32)$, which checkpoint to transfer $(1000, 2000, 3000)$, as well as the fine-tuning learning rate $(10^{-4}, 5 \times 10^{-5}, 10^{-5})$ and batch size $(8, 32)$. We find that for all three source corpora (**es**, **ec**, **paren-zipf**), it works best to pre-train with learning rate $5 \times 10^{-4}$ and batch size $(32)$, transfer using the checkpoint with 3000 training steps, and fine-tune with learning rate $10^{-4}$ and batch size 8. For training from scratch, we have tried grid search for the learning rate $(10^{-3}, 5 \times 10^{-4}, 10^{-4}, 5 \times 10^{-5})$ and batch size $(4, 32)$, and find that learning rate $10^{-4}$ and batch size 8 work best for different downstream languages. An pre-training experiment can finish within one hour using one GeForce RTX 3090 GPU, while a fine-tuning or training-from-scratch experiment can finish within one hour using one GeForce RTX 2080 GPU.

## A.3    IMAGE CAPTIONING

We use the pre-processed detection features[9] of Conceptual Captions from the codebase of Li et al. (2020a).

For both pre-training and fine-tuning, we use a public codebase[10] for image captioning based on FAIRSEQ (Ott et al., 2019), and mostly follow their default setups. Pre-training on Conceptual Captions takes 8 GeForce RTX 3090 GPU for around two days. Fine-tuning takes 1 GeForce RTX 2080 GPU for one hour.

# B    ADDITIONAL RESULTS

## B.1    LANGUAGE UNIGRAMS

As shown in Figure 4, the **es** and **paren-zipf** corpora have a larger vocabulary size (5000) and a larger entropy (6.48). While **ec** is set with vocabulary limit 4,035, its corpus only uses around 2,500 words with smaller entropy (3.7). The **ec** corpus with **random speaker** almost has a large entropy (7.98).

---

[6] https://github.com/cambridgeltl/ECNMT/tree/master/ECPRETRAIN
[7] https://github.com/toizzy/tilt-transfer/tree/master/corpora/create_wiki_corpus
[8] https://github.com/huggingface/transformers/blob/v4.4.2/examples/language-modeling/run_clm.py
[9] https://github.com/microsoft/Oscar/blob/master/VinVL_DOWNLOAD.md
[10] https://github.com/krasserm/fairseq-image-captioning

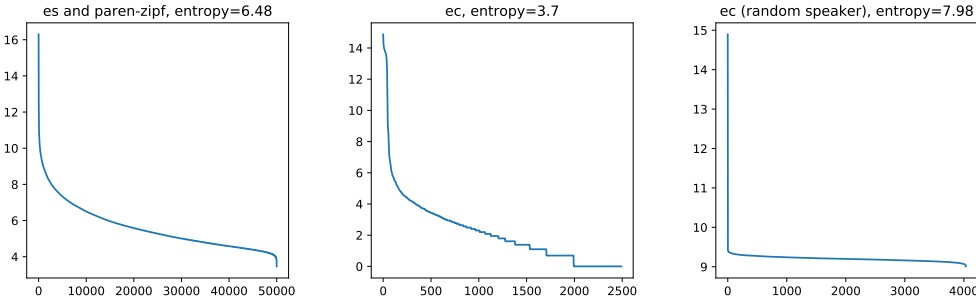

Figure 4: Unigram distributions of (1) **es** and **paren-zipf**, (2) **ec**, and (3) **ec** with **random speaker**.

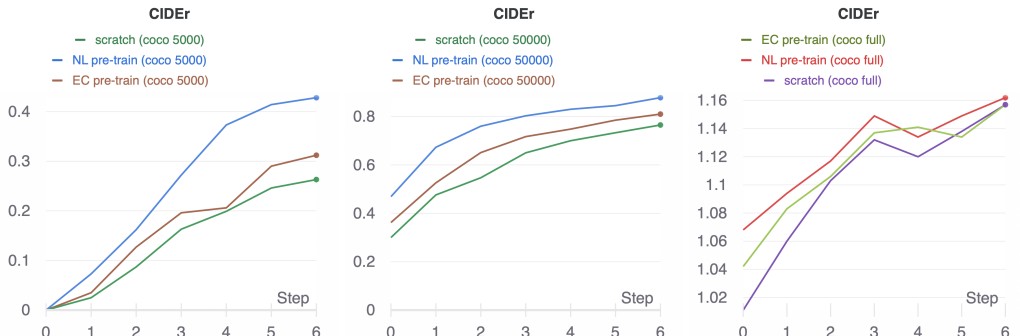

Figure 5: The validation CIDEr (Vedantam et al., 2015) score across different fine-tuning epochs, when using 5,000, 50,000, or the all samples of MS-COCO training samples.

### B.2 IMAGE CAPTIONING

We visualize the fine-tuning process of image captioning experiments in Figure 5. Interestingly, we find that under different natural language resource conditions (5,000, 50,000, or all samples in the MS-COCO (Lin et al., 2014) training set) the training progress is different. Specifically, with 5,000 samples, EC or NL pre-training and training from scratch first learn similarly well, then gaps gradually appear with more training epochs. In contrast, when more han 50,0000 samples are used, the gap between pre-training methods and training from scratch is most significant when trained for only one epoch, and it starts to diminish with more training epochs. It suggests that even when downstream natural language resources are abundant, pre-training on an EC corpus might still help in a fast adaption setup.

We also perform additional ablation studies to confirm the non-triviality of image captioning results. Due to computation limits, we use 450k MS-COCO samples (instead of 3M Conceptual Captions samples used in the main paper) to pre-train (image $\mapsto$ emergent caption / shuffled emergent caption / random paren-zipf string with the same unigram as emergent captions), and the other 50k MS-COCO samples to finetune (image $\mapsto$ English caption). As shown in Table 4, pre-training with shuffled emergent captions or ungrounded strings leads to worse performance than even no pre-training, while pre-training with emergent captions still improves the downstream performance. This helps address the importance of having a semantically grounded and structural language for vision-language pre-training. However, we do note that in the case of MS-COCO image captioning, fine-tuning usually does the heavy-lifting and pre-training contributes much less than the case of language modeling. That is why our work focuses analysis on the language modeling experiments.

### B.3 CORRELATIONS OF METRICS AND DOWNSTREAM PERFORMANCE

Figure 6 plots the 200 data points in terms of their metric values and downstream performance on Romanian and Hebrew modeling, respectively. We note that topographic similarity is usually very

| Pre-training language | CIDEr |
|---|---|
| Emergent language | **62.5 (1.4)** |
| Emergent language (shuffled) | 59.2 (0.5) |
| Paren-zipf | 59.5 (0.2) |
| No pre-training | 60.9 (0.5) |

Table 4: Image captioning results with different pre-training languages or no pre-training.

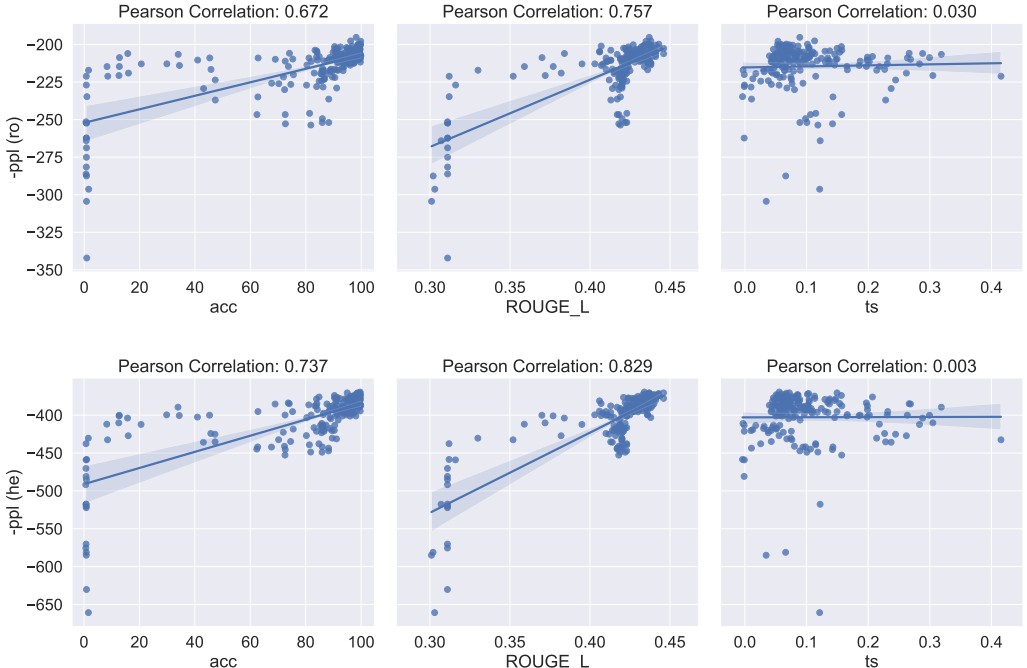

Figure 6: 200 data points with x axis being metrics (validation acuracy, translation ROUGE_L, topographic) and y axis being downstream performance (negated perplexity) on Romanian (ro) and Hebrew (he), respectively.

low (less than 0.1), and a higher value does not correlate with better downstream performance. Also, we note that data points around the beginning of training (e.g. 200 steps) have very low accuracy or translation score, and their downstream performance is worse but significantly more varied than the rest of data points.

### B.4    HOW SETUPS (VOCABULARY SIZE, SEQUENCE LENGTH) EFFECT METRICS AND DOWNSTREAM PERFORMANCES

Figure 3 in Section 5 mainly studies how metrics and downstream performance change with respect to the training steps, while controlling the other setups. Following a similar logic, in Figure 7 we take those checkpoints with 1000 training steps and sequence lengths being 15 and plot the downstream performances and metrics with respect to the vocabulary size (1000, 4035, 10000), and in Figure 8 we take those checkpoints with 1000 training steps and vocabulary size being 4035 and plot the downstream performances and metrics with respect to the sequence length (5, 15, 25).

For the vocabulary size setup (Figure 7), we find that both validation accuracy and our metric capture that the performance when vocabulary size is 1000 is worse and has significantly more variance. In contrast, topographic similarity is highest when vocabulary size is smallest (1000), which disagrees with downstream performances. However, all metrics are higher when vocabulary size is 4035 instead of 10000, which is the opposite for downstream performances.

For the sequence length setup (Figure 8), we find that our metric captures the fact that a larger sequence length leads to better and less varied downstream performance. In contrast, validation accuracy with sequence length being 25 is worse than sequence length being 15, and topographic similarity fails to capture the decreasing trend of performance variance.

In summary, we find that a larger vocabulary size or sequence length within our setup choices leads to better downstream performance with less variance, which is best captured by our translation metric.

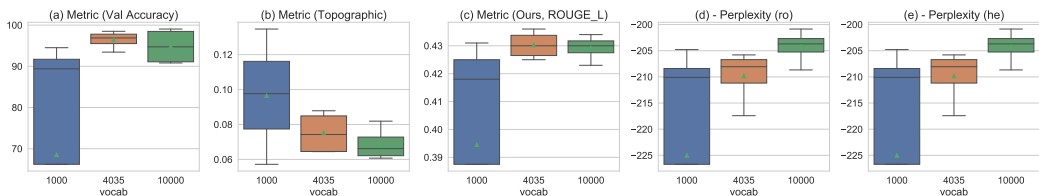

Figure 7: How metrics and downstream performances change with respect to vocabulary size (1000, 4035, 10000). Sequence length is 15 and training step is 1000.

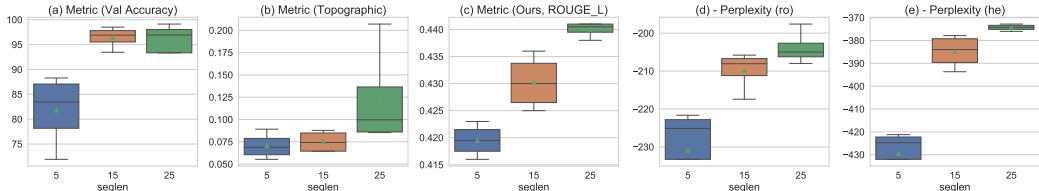

Figure 8: How metrics and downstream performances change with respect to sequence length (5, 15, 25). Vocabulary size is 4035 and training step is 1000.

## B.5 QUALITATIVE EXAMPLES

Figure 9 includes some COCO images (not Conceptual Captions training images for EC) and their natural and emergent captions. Manual check might reveal some patterns, e.g. transportation tools might start with 2340 (a, b, j), indoor scenes might start with two 1777 tokens and indoor home scenes might start with even more 1777 tokens (d, e, k, l). Still, we note that our paper intentionally aims to move away from previous manual-check paradigms in EC papers and tries to establish a more standardized, scalable, and robust quantitative metric to enable more progress.

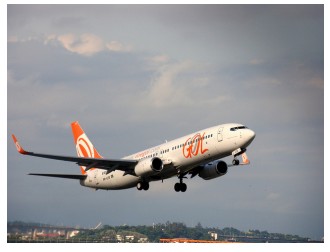

(a) large passenger airplane flying through the air. [2340, 2403, 2308, 320, 1329, 2308, 2308, 1329, 3643, 1512, 4033, 1298, 1526, 1526, 0]

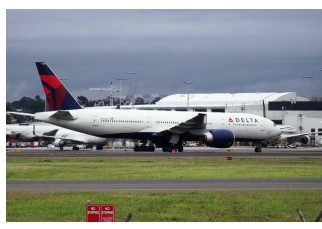

(b) delta airplane at the airport on the runway. [2340, 2422, 2403, 141, 2422, 320, 2340, 2308, 3701, 110, 3701, 397, 1241, 587, 0]

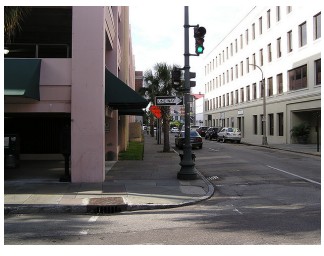

(c) green street light in between two buildings. [1603, 320, 2422, 1965, 2403, 3098, 319, 1965, 1307, 1526, 1307, 2389, 1526, 1223, 0]

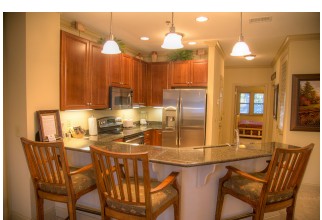

(d) this kitchen has stainless steel appliances and granite countertops. [1777, 1777, 1777, 1777, 1298, 1777, 3682, 2823, 3701, 3701, 3643, 4033, 1526, 1526, 0]

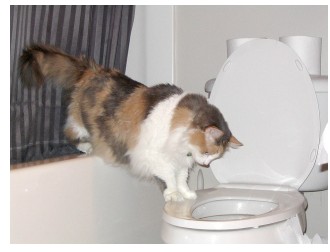

(e) black white and brown cat looking into a toilet bowl. [1777, 1777, 1777, 1043, 1777, 1043, 1976, 1777, 2422, 2550, 2762, 477, 2422, 3715, 0]

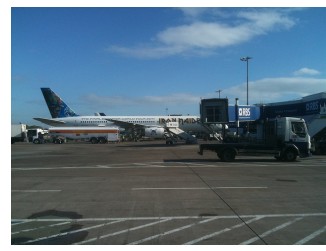

(f) large airplane and a truck next to a building. [1777, 2403, 2422, 2308, 1965, 814, 2308, 397, 1526, 477, 2389, 1526, 2389, 1526, 0]

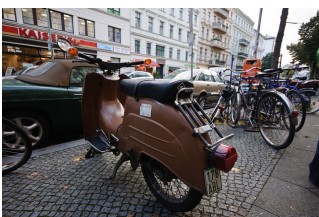

(g) there are bicycles parked along a stone sidewalk. [1777, 1965, 2403, 1329, 1329, 1329, 319, 2277, 1329, 1526, 1526, 3701, 1526, 2277, 0]

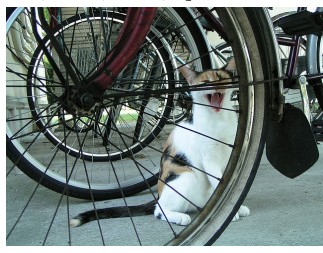

(h) cat sitting next to a bunch of bikes parked next to each other. [2422, 1777, 1043, 320, 1329, 1043, 3643, 3682, 3643, 477, 3701, 2944, 2277, 2389, 0]

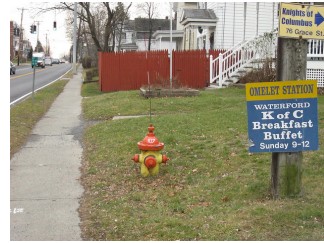

(i) photo of a sidewalk and yard with a fire hydrant and signs. [1603, 2422, 319, 1603, 1043, 319, 2762, 320, 2277, 2762, 141, 3701, 2389, 1526, 0]

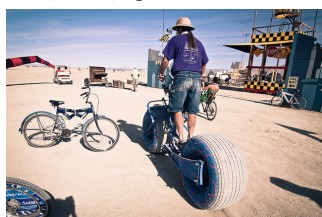

(j) man standing on a vehicle with two large wheels. [2340, 1777, 1965, 319, 1329, 3682, 1329, 1329, 3643, 1329, 165, 3643, 2389, 2389, 0]

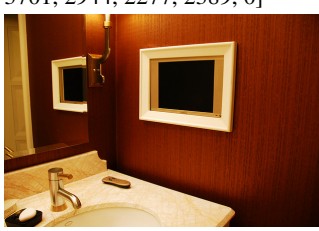

(k) television screen is mounted in the wall of the bathroom. [1777, 1777, 1777, 1777, 1777, 2762, 1965, 2550, 3226, 4012, 2823, 2550, 2655, 4033, 0]

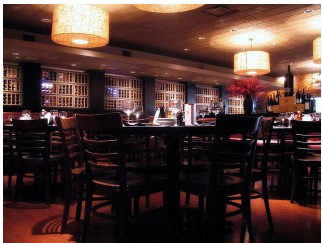

(l) restaurant that has many tables set up . [1777, 1777, 3098, 320, 320, 1298, 1965, 2823, 3701, 1298, 1714, 1241, 3701, 4033, 0]

Figure 9: Images from COCO and their natural and emergent captions.

