# OpenReview forum: "Linking Emergent and Natural Languages via Corpus Transfer"
_ICLR.cc/2022/Conference — ICLR 2022 Spotlight_

### Official Review · Reviewer_gtt4 · 2021-10-28

**Correctness:** 3
**Technical Novelty And Significance:** 3
**Empirical Novelty And Significance:** 3
**Recommendation:** 8
**Confidence:** 4

**Main Review:**

# Summary

The authors propose to use corpora generated from _emergent communication_ as a fine-tuning signal for NLP tasks (language modeling and image captioning in particular).  They show that, especially in small-data regimes, pre-training on an emergent language can yield significant performance boosts in both tasks.  (In particular, pre-training on emergent language performs on average better than synthetic hierarchical data, but not quite as well as a different natural language, and all of these pre-training methods do better than training from scratch.)  This is the newest of a small but growing body of literature that seeks to connect emergent communication with genuine NLP tasks.  The results are intriguing and promising and should be of interest both to the emergent communication community as well as to the broader community working on low-resource NLP.

Strengths:
* Novel method of using emergent language for pre-training (as opposed to transferring an entire artificial agent)
* Some good ablations to identify what contributes to successful transfer
* A new evaluation metric (emergent --> NL translation performance) that best correlates with fine-tuning performance

Weaknesses:
* Some parameter choices and the design of some ablations are not completely justified
* Some additional related works could be included


# Minor comments / questions

* "However, this metric is too rigid in its definition of compositionality, ignoring aspects like argument structure, context or morphology which play a key role in determining the combination of word semantics (Goldberg, 2015)."  Steinert-Threlkeld (2020) "Towards the Emergence of Non-trivial Compositionality" makes similar points and could be cited here as well.

* Why are the lines for "from scratch" flat in Figure 2?  I would have thought this was training an LM on progressively larger portions of the relevant data (being used for fine-tuning the others), in which case I'd also expect a downward trend in perplexity.  Are these rather the ppl resulting from training an LM on the full dataset?  It should be stated more clearly in the text what this means.

* Why does the EC pre-training use |V| = 4035, as opposed to 50 that's used in the other tasks and the fine-tuning corpora?  Do the authors expect better results with a larger vocab size?

* Ablation: the model vs corpus transfer comparison seems unfair to me.  In particular, you are comparing a small GRU LM to a larger transformer LM, where the latter is, as you mention, a much more powerful model.  While it is also true that corpus transfer _enables_ this divergence in model types, to really test whether transferring the whole model versus using the corpus works or not, I would think you would want to compare fine-tuning the sender GRU on the LM data vs. starting from scratch _with a GRU of the same type_ and pre-training on the EC corpus before fine-tuning.  Did you do an experiment of that type?

* Related work: I would also include a discussion of this Lazaridou et al paper where they compare ways of combining EC with non-EC learning signals (e.g. image caption training): https://aclanthology.org/2020.acl-main.685.pdf

* Ethics statement: I appreciate this statement and agree with the possible positive impacts.  Do the authors see any potential negative impacts?  If not, that should also be explicitly stated.


# Typographic comments

* p 1, "the input out of detractors" --> "the input out of distractors"

* p 2: "transferable benefits for downstream natural language tasks" the single hyphens surrounding the subsequent list should be em dashes (three hyphens in TeX)

* p 3: "uses another GRU layer to decode the message m into a hidden vector hl" I would use "encode" instead of "decode" here, since text-->representation is usually what an encoder does

* p 3: "The most straightforward metric is the accuracy of playing the referential game p(guess = Ii)."  Given the way p(guess=Ii) is used above, I think this should be more like E[argmax(p(guess=Ii)) = i].  Or if they are measuring the probability assigned to the true image and not just accuracy, the name shoudl be changed from accuracy.

* p 4: "es should set a upper bound" --> "es should set an upper bound"

* p 5: " ec perform better than or " --> " ec performs better than or "

**Summary Of The Paper:**

The authors propose to use corpora generated from _emergent communication_ as a fine-tuning signal for NLP tasks (language modeling and image captioning in particular).  They show that, especially in small-data regimes, pre-training on an emergent language can yield significant performance boosts in both tasks.  (In particular, pre-training on emergent language performs on average better than synthetic hierarchical data, but not quite as well as a different natural language, and all of these pre-training methods do better than training from scratch.)  This is the newest of a small but growing body of literature that seeks to connect emergent communication with genuine NLP tasks.  The results are intriguing and promising and should be of interest both to the emergent communication community as well as to the broader community working on low-resource NLP.

**Summary Of The Review:**

The paper presents a novel method and evaluation of using emergent communication for pre-training models on real NLP tasks.  This has the potential to help in low-resource settings, and demonstrates the value of emergent communication for NLP in addition to theoretical interest in language evolution.

---

> ### Author Response · Authors · 2021-11-12
> **Response to Reviewer gtt4**
>
>
> Thank you for finding our work "intriguing and promising and should be of interest both to the emergent communication community as well as to the broader community working on low-resource NLP".
>
> 1) **lines for "from scratch" flat in Figure 2**
>
> We are sorry this is confusing. Training from scratch (red line) is a flat line because there is no pretraining corpus, so it is just one single value. We will revise the line to be dotted and better explain in revision.
>
> 2) **EC vocab**
>
> Vocabulary size of 4035 follows the default setup from Li et al. (2020). Prior work that uses a vocabulary size as small as 50 usually operates in simpler and smaller domains. In section 4 we do consider varying vocabulary sizes (1000, 4035, 10000), and we will add more results about how setups affect metrics and downstream performances in the appendix. For example, taking those checkpoints with training steps = 800 and sequence limits = 15, downstream performances (lower is better) and metrics (higher is better) are shown below. We find **a small vocab size (1000) slightly worsens average performance, but makes variance significantly higher**. Vocabulary sizes 4035 and 10000 are quite similar. Such trends are captured by our translation metric and validation accuracy (though accuracy when vocab=1000 is a bit too low in light of the downstream performance), but not topographic similarity, which is consistent with our analysis in Figure 3 about training steps.
>
> |Vocab size  | LM (ro) | LM (he) |  Translation ROUGE_L| Valid acc |  Topographic |
> | -------- | -------- | -------- | -------- | -------- | -------- |
> | 1000   | 222 (27)   |  425 (56)    | 39.6 (4.9) | 67.0 (38.4) | 0.11 (0.04) |
> | 4035    |  209 (3)     |    385 (6) |  43.3 (0.4)| 94.8 (1.8) | 0.09 (0.01) |
> | 10000    |  211 (2)     |    393 (3) | 43.2 (0.6) | 94.6 (4.7) | 0.08 (0.01) |
>
>
> 3) **the model vs corpus transfer comparison**
>
> We believe this is a very good point, and we address the broader issue (corpus vs. model transfer) in General Response (2). To be more concrete to your questions:
> - We conduct new experiments that show **corpus transfer outperforms model transfer even with just a one-layer GRU**, possibly due to the "self-distillation" benefit (e.g. https://arxiv.org/pdf/2012.09816.pdf).
>     | Transfer (LM model) | LM (ro) | LM (he) |
>     | -------- | -------- | -------- |
>     | corpus transfer (Transformer)   |  198 (4)    |  375 (14)    |
>     | **corpus transfer (1-layer GRU)**   |  553 (53)       | 1056 (55)  |
>     | model transfer (1-layer GRU)    |  655 (40)        |     1105 (6) |
> - Unfortunately, unlike most NLP tasks, **architectures for EC agents might not be arbitrarily big/expressive**. For example, we have tried changing EC model GRUs from 1 to 2 layers, which already led to worse game accuracy and corpus transfer performance (see below), possibly because optimization with Gumbel-softmax (or REINFORCE) becomes harder. We believe Transformers as EC models will be even harder, and to our knowledge, this has not been tried by prior work before.
>     | EC speaker model | LM (ro) | LM (he) |
>     | -------- | -------- | -------- |
>     | 1-layer GRU  |  198 (4)    |  375 (14)    |
>     | **2-layer GRU**   |  249 (54)       | 506 (157)  |
> - While our current draft highlights the practical advantage of corpus transfer over model transfer, they also bear very different motivations and scientific focuses. Please see General Response (2) in more detail.
> - We will update Table 2 with new results and clarify the architecture of different rows.
>
>
>
> 4) **Related work and typos**
>
> Thank you for the detailed comments! We will address the suggestions in revision.
>
> 5) **Ethics statement**
>
> We cannot think of any negative ethics implications for now, given the emergent language is still quite different from natural language. We are happy to discuss further on this.

---

> ### Author Response · Authors · 2021-11-18
> **Response to Reviewer gtt4: Revision Updated**
>
> Dear Reviewer gtt4,
>
> Thanks again for your detailed feedback, according to which we have revised the paper significantly:
>
>
> 1. **Analysis of additional EC setups (vocab size, sequence length) (Appendix B.4)**. We really appreciate your detailed question about EC vocab size. As a result, we added in Appendix Section B.4 how vocabulary size and sequence length effect different metrics and downstream performances, with new results to support that larger vocabulary size/sequence length could lead to better downstream performance with less variance, which is best captured by our translation metric (over validation accuracy and topographic similarity). Let us know if this addressed your question about vocabulary size.
> 2. **Better comparison to model transfer**. We added the point that (i) transfer is also used in our work as a means to understanding and analyzing properties of emergent language beyond the game framework (Section 1), (ii) emergent language is a  representation with more general properties beyond parameters of particular EC models (Section 3.1), and (iii) more detailed discussion of significant practical advantages of corpus transfer over model transfer (Section 4.3), with added experiment results to confirm such an advantage even when corpus transfer uses the 1-layer GRU, and the clarficiation about the scalability of EC model architecture, thanks to your great suggestions. Finally, we summerize those points in Section 2 Related Work for a more comprehensive comparison to Li et al. (2020).
> 7. **Table 2 update**. We have updated Table 2 with standard deviations, and used dotted red lines for training from scratch with text explanations. Thanks again.
> 8. **Related Work**. We cited Steinert-Threlkeld (2020) in Section 1 and 5 when discussing the measure of NL compositionality, and Lazaridou et al. (2020) in Section 2 and 4 when discussing EC/NL mixing training.
> 9. **Typos**. We have fixed all typos you pointed out. Thank you!
>
> *We sincerely appreciate your comments. Please let us know if you have further feedback.*
>
> Best, Authors

---

### Official Review · Reviewer_QkKj · 2021-11-01

**Correctness:** 3
**Technical Novelty And Significance:** 3
**Empirical Novelty And Significance:** 3
**Recommendation:** 6
**Confidence:** 3

**Main Review:**

Strengths:
- I think this research question is important and interesting, and that the study of what EC agents output and how they might relate to natural language is important for both EC and NLP
- The setting was made very clear. Figure 1 is excellent.
- The claims, particularly those about language modeling, are evaluated on a variety of settings

Weaknesses:
- Some of the claims don't seem entirely justifiable. It's true that pretraining on EC is better than pre-training on ES or parens at 2M tokens, but it's not entirely clear to me that this is an interesting setting because _all_ models perform poorly. It's more interesting to me that ES is similar to parens
- It's not clear to me how specific to this particular experimental setup the proposed evaluation technique is. It seems to correlate better with LM performance in this setting, but how about others? Using a particular method for evaluation requires some degree of confidence that it holds generally, and I'm not sure that the results here convince me of that.

**Summary Of The Paper:**

This paper studies whether output from emergent communication systems might be useful as pre-training data for natural language tasks. The authors train an EC agent, generate from it, and use the generated IDs as data for language model pre-training. Compared to pre-training on Spanish wikipedia or a simple bracket language, pre-training on EC-generated data is better when you have 2M examples, but quickly becomes worse as the amount of data increases. In experiments on image captioning, the authors find that pre-training on either EC-generated or natural language captions improve over not pre-training, but that there is not much of a difference between the results when pre-training on EC-generated text and natural language. Finally, the authors propose EC -> natural language translation as a way of evaluating the quality of EC agents.

**Summary Of The Review:**

I think this work addresses an interesting question, but I disagree with a few of their claims based off of the experimental evidence provided. The broader research question is important enough that I feel like even initial work in this direction is perhaps of interest to the ICLR community and worth accepting.

---

> ### Author Response · Authors · 2021-11-12
> **Response to Reviewer Qkkj**
>
>
> Thank you for finding our research questions "interesting" and "important for both EC and NLP". Please refer to General Responses and our responses to other reviewers that provide a more comprehensive view of our work.
>
> 1) **Some of the claims don't seem entirely justifiable**
>
> We are unsure exactly which claims you refer to but will be happy to address accordingly once we know.
>
> 2) **pre-training on ES or parens at 2M tokens is not interesting**
> - We note that this direction of leveraging EC for NLP tasks is very recent, and **we are the first work to show any evidence that under the same corpus size, an emergent language corpus could provide greater transfer benefit compared to a natural language**. Such a finding, despite being preliminary, is **very surprising in the context of prior work** (Li et al. (2020) do not compare to NL pretraining, while in Papadimitriou & Jurafsky (2020) no artificial languages (music, code, regular languages) could transfer better than natural languages), and should be interesting and relevant for both EC and NLP communities.
> - All models perform poorly with 2M fine-tuning tokens because we are explicitly trying to simulate a **low-resource setup** that resembles intended applications for emergent language pre-training where natural language annotations are truly scarce (e.g. low-resource languages of which even unlabeled corpus is limited, embodied tasks that are hard to annotate).
> - As mentioned in the "Limitations and Future Directions" part of our paper, our methodology and findings could be a starting point to inspire future work that pushes EC to be more useful for NLP and ML. Please refer to our Response to Reviewer wvqW (3) for more details.
>
>
> 3) **"ES is similar to parens" is more interesting**
>
> By "ES is similar to parens", do you mean their trends with respect to pretraining corpus size are similar despite their consistent gaps? We note that training from scratch (red line) is meant to be a flat line (because there is no pretraining corpus; we will revise the line to be dotted and better explain), and **the trends of ES and paren-zipf (i.e. pre-training only bootstraps with big enough corpus size) are arguably expected to be true for all pre-training languages**. To us, it is more surprising and interesting how pre-training with emergent language can be bootstrapped with as little as 2M tokens, and we don't know any other pre-training languages with such a trend (pre-training bootstraps with small corpus but hard to scale). Let us know if we misinterpret anything.
>
> 4) **correlating better with LM performance in this setting is particular**
>
> - We are not entirely sure what "this particular experimental setup" and "others" exactly refer to. We will be happy to address more concrete comments.
> - Our translation metric links emergent languages to English (IE-Germanic) image captions, while the language modeling uses Wikipedia corpora of two different natural languages, Romanian (IE-Romance) and Hebrew (Afro-Asiatic). Even though the metric and downstream tasks involve very different content and linguistic features, the correlation is shown to be consistent. We are happy to include more languages if needed.
> - We plan to add in the appendix more details about the 200 trials from Section 4, and how sequence length/vocabulary size/training steps affect different metrics and downstream performances. For a concrete example, see Response to Reviewer gtt4 (2), and how our metric also correlates with downstream performance in such a more controlled setup.

---

> ### Author Response · Authors · 2021-11-18
> **Response to Reviewer QkKj: Revision Updated**
>
> Dear Reviewer QkKj,
>
> Thanks again for your comments.
>
> Today we have made substantial changes in the revision. In particular, we added in Appendix Section B.4 how vocabulary size and sequence length effect different metrics and downstream performances as promised. We find that larger vocabulary size/sequence length could lead to better downstream performance with less variance, which is best captured by our translation metric (over validation accuracy and topographic similarity). We appreciate your comment and believe the added results make our findings more robust and less particular.
>
> *Please refer to **General Response: Revision Updated** for more revision changes, and let us know if you have any more questions.*
>
>
> Best, Authors

---

### Official Review · Reviewer_G942 · 2021-11-02

**Correctness:** 3
**Technical Novelty And Significance:** 4
**Empirical Novelty And Significance:** 4
**Recommendation:** 8
**Confidence:** 4

**Main Review:**

# Strengths

- This is a great idea, and one of those ideas that I (and others) will wish they had come up with. The idea of pretraining on emergent corpora neatly brings together new ideas about transfer learning and modern studies of emergent communication.
- Significant experiments in both grounded (captioning) and ungrounded (LM) settings convincingly demonstrate the efficiency of EC pretraining. The ablation studies are also very useful, but also open up several questions (see Weaknesses).
- There is a very rich space of future experiments to try in this area. I imagine it will inspire plenty of follow-up work, exploring how different aspects of the EC corpus do or do not result in better transfer. This may also become a typical evaluation w.r.t. how useful/systematic a language corpus is—how useful is it for downstream transfer?
- The natural language translation metric is extremely clever, and is a natural extension of similar ideas for "translating" emergent sentences (Andreas et al., 2018).

# Weaknesses

- Poor comparison to parameter transfer approaches
    - Perhaps the biggest issue I have with the paper is the lack of comparison to parameter transfer approaches (Li et al., 2020). The GRU transfer line in Table 2 is the right idea, but is woefully insufficient: it makes no sense to compare a 1-layer GRU perplexity to transformer complexity, and there's no reason that you can't generate the EC corpus with transformers and therefore more cleanly compare corpus transfer to parameter transfer. I would be really interested in these results. To be clear, I don't think it's necessary that corpus transfer has to do "better" than parameter transfer for the paper to be useful—there are some potential advantages of corpus transfer: (1) learning from EC corpora generated by heterogenous models, (2) cheaper to generate EC corpora from a cheaper model. But I still really want to see this comparison.
    - I don't think this is a fatal weakness, and perhaps it illustrates follow up experiments that this work inspires, which is to vary the architecture/capability of the EC agents and seeing what kinds of agents produce EC corpora that are more useful for transfer, and how those EC corpora differ using the various tools we have for analyzing emergent languages (e.g. topographic similarity).
    - In fact, I think this experiment is nonsensical and misleading enough (every other line in the table uses a Transformer), that, without rerunning modified experiments, I would encourage authors to just remove this number entirely.
- Limited evaluation of non-EC pretraining for image captioning.
    - The experimental evaluation is overall quite comprehensive, except in the image captioning studies. Authors say "We note that such a transfer scheme cannot work for other synthetic corpora such as music, code, or paren-zipf as they are not grounded on perceptual stimuli." While these other corpora indeed are not grounded in perceptual stimuli, you could still imagine pretraining on the ungrounded data, then finetuning while introducing grounding, or even arbitrarily associating different strings from a corpus with images. I would really like to see this comparison, as part of the strengths of Sec 3.1 are in the careful comparison of EC pretraining w/ other choices of pretraining corpora.
- Results on permuted EC corpora could be explained further
    - One outstanding question-issue I have with this paper is the results on permuted EC corpora. As I understand it, this is for the ungrounded language modeling task for ro and he languages. The permuted EC corpus does not seem substantially worse than EC pretraining (e.g. 195 vs 211 perplexity is not a huge difference compared with 211 vs 266), so I'm somewhat suspicious of the author's claim that "speaker develops a coding scheme with sentence structures beyond bag of words."  A significance test would be helpful to measure the effect here. But regardless, it's clear that training on permuted EC results in impressive gains. Then what benefit does the EC corpus actually provide? The permutation experiments show that it's not syntax/sequential structure. And it's not learning a grounding between objects in images and emergent language tokens, since these numbers are for the ungrounded LM task. Maybe it's just token colocation statistics then?
    - As an aside, I would really love to see the same experiments made with the image captioning task as well (e.g. even if the language ignores sequential structure, the bag of words annotation of relevant objects/features in the grounded input that the permuted EC corpus provides may be just as good as the original EC corpus)
- Connections to related work.
    - There have been a few attempts ta combining emergent communication and supervision, e.g. via multitask training ([Lowe et al., 2021](https://arxiv.org/abs/2002.01093), also [Lazaridou et al., 2020](https://arxiv.org/abs/2005.07064)) which are missing from related work. The connection could be made more clear. Why might we expect corpus transfer to do better or worse than multitask training? Perhaps these methods could be combined, e.g. multitask training on EC corpus and real corpus?

# Questions/Minor

- How does changing the vocabulary size and bandwidth of the EC pretraining corpus change its utility for pretraining? There is surely a point at which EC corpora are too simple (e.g., imagine just a single token and a very small vocab). Would be interesting to quantify the benefits gained not just by varying corpus size, but by varying emergent language complexity. Figure 3 partially answers this question (in that untrained EC corpora are not helpful for transfer), but I'd be interested, for example, in seeing a plot of (V, T) and transfer efficacy.
- One of the central issues in studies of emergent communication is that agents often develop non-compositional, unintuitive communication protocols. The EC setup described in 2.1 doesn't seem to have any of the guards against degenerate languages that more modern studies of EC have investigated, e.g. different ImageNet categories (Lazaridou et al., 2017) or SimCLR views ([Dessi et al., 2021](https://arxiv.org/abs/2106.04258)), or even other ways of regularizing a language (e.g. [Luna et al., 2020](https://arxiv.org/abs/2004.03868)) Yet the EC languages still seem useful for transfer. So questions are, (1) do we still see degeneracy in the languages used in the EC corpora, and (2) if so, does trying to reduce degeneracy/improve systematicity of the language result in better transfer performance?
- Why are there no standard deviations for EC pretrain/from scratch in Table 2?
- Table 3 - how many data points actually go into the computation of correlations here? Is it 5 x 4 = 20? I would really love to see the full scatterplots rather than just correlation, e.g. in the appendix, and to actually see how much variance we see in the different EC corpora, as measured by the metrics.
- The details of the training steps expt in Figure 3 are a little underexplained, and I tried my best to reconstruct, though I invite the authors to clarify if I've misunderstood anything. You take checkpoints of the EC corpora generated from EC agents at 0, 200, 400, ... 1k training steps, and at each step measure the various EC langauge metrics, and also do the pretrain/finetune experiment with transformers, starting with the EC corpora, to get plot d - perplexity. Is that correct?

**Summary Of The Paper:**

This paper proposes a new way of using techniques from the emergent communication literature, where agents are trained to develop languages for communication on some shared task, to improve more typical supervised learning tasks in NLP, namely language modeling and image captioning. This is a question that is only just beginning to be addressed in some work (e.g. Lowe et al., 2021; Lazaridou et al., 2020), so this area is ripe for exploration. This paper proposes an interesting and novel method of using EC to improve supervised learning: generate emergent language from an image-based referential game with simple speaker/listener models. Then, using a much larger transformer, pretrain on this corpus before doing fine-tuning on the supervised task of interest, a la Papadimitriou and Jurfasky, 2020.

Both language modeling and image captioning tasks are explored, and results convincingly show that emergent communication corpora are surprisingly effective. Some ablation studies help elucidate where the benefits arise, although there are some questions here (see Weaknesses).

Finally the authors propose a new metric for evaluating the naturalness and usefulness of a corpus of emergent language: if you have natural language messages generated for some task, as well as true human messages, the authors literally just build a model for translating emergent languages into the corresponding human messages. The ROUGE score obtained by such a model is thus a "transferability" metric. This metric is not only useful for identifying whether an EC corpus will be useful for transfer for some task (more similar = more useful), but is possibly also a measure of humanness/compositionaliy of a language in general, and so this contribution will be useful even for those in EC who are not interested in supervised learning.

Overall, I very much like the philosophy of this paper. I think it helps fill an important gap in the literature, which is namely what to do with the recent deluge of emergent communication studies. The big elephant in the room with these studies is whether the languages are actually useful for some reason besides being studies of linguistic evolution with limited generality. This paper proposes a simple and interesting way of using the emergent languages for something more productive and more of interest to people actually working on real NLP applications. I do however see some weaknesses and lack of comparison which I would love to see answered and/or addressed in the final version.

**Summary Of The Review:**

Overall, I think this is a creative, technically sound, and very interesting paper with valuable contributions to the emergent communication literature (notably, the big question of how to make EC useful for the rest of the community), and the paper really made me think. It's a shame that there are some unanswered questions remaining in the paper (see Weaknesses). I hope the authors might consider some of these follow-up questions and experiments during the rebuttal phase, but I think even despite these flaws, the paper has enough contributions to stand on its own. If all of the issues in the weaknesses section are addressed I would consider raising my score to a 10.

---

> ### Author Response · Authors · 2021-11-12
> **Response to Reviewer G942**
>
>
> We are glad that you "very much like the philosophy of this paper" and find our work "fill an important gap in the literature". **Here we address "all of the issues in the weaknesses section" with new experiments and results**.
>
> 1) **Poor comparison to parameter transfer approaches**
>
> We believe this is a very good point, and we address the broader issue (corpus vs. model transfer) in General Response (2). To be more concrete to your questions:
> - We conduct new experiments that show **corpus transfer outperforms model transfer even with just a one-layer GRU**, possibly due to the "self-distillation" benefit (e.g. https://arxiv.org/pdf/2012.09816.pdf). Due to the large variance of 1-layer GRU language modeling, we run 6 trials for corpus and model transfer with 1-layer GRU to confirm the gap is significant.
>     | Transfer (LM model) | LM (ro) | LM (he) |
>     | -------- | -------- | -------- |
>     | corpus transfer (Transformer)   |  198 (4)    |  375 (14)    |
>     | **corpus transfer (1-layer GRU)**   |  553 (53)       | 1056 (55)  |
>     | model transfer (1-layer GRU)    |  655 (40)        |     1105 (6) |
> - We note that, unlike most NLP tasks, **architectures for EC agents might not be arbitrarily big/expressive**. For example, we have tried changing EC model GRUs from 1 to 2 layers, which already led to worse and less stable corpus transfer performances (see below), possibly because optimization with Gumbel-softmax (or REINFORCE) becomes harder. We believe Transformers as EC models will be even harder, and to our knowledge, this has not been attempted by any previous work and might be a potentially important future direction.
>     | EC speaker model | LM (ro) | LM (he) |
>     | -------- | -------- | -------- |
>     | 1-layer GRU  |  198 (4)    |  375 (14)    |
>     | **2-layer GRU**   |  249 (54)       | 506 (157)  |
> - While our current draft highlights the practical advantage of corpus transfer over model transfer, they also bear very different motivations and scientific focuses. Please see General Response (2) in more detail.
> - We will update Table 2 with new results and clarify the language model architectures of different rows.
>
>
> 2) **vary the architecture/capability of the EC agents**
>
> This is also a good suggestion, and we plan to add in the appendix more details about the 200 trials from Section 4, and how sequence length/vocabulary size/training steps affect different metrics and downstream performances. For a concrete example, see Response to Reviewer QkKj (2).
>
>
> 3) **more pre-training languages (permutated EC, paren-Zipf) for image captioning**
>
> Due to computation and time limits, we use 450k MS-COCO samples (instead of 3M Conceptual Captions samples used in Section 3) to pre-train (image -> emergent caption / shuffled emergent caption / random paren-zipf string with the same unigram as emergent captions), and the other 50k MS-COCO samples to finetune (image -> English caption), and the results are below. We find that **pre-training with shuffled emergent captions or ungrounded strings leads to worse performance than even no pre-training, while pre-training with emergent captions still improves the downstream performance**. This helps address the importance of having a semantically grounded and structural language for vision-language pre-training. However, we do note that in the case of MS-COCO image captioning, fine-tuning usually does the heavy-lifting and pre-training contributes much less than the case of language modeling. That is why our work focuses analysis on the language modeling experiments.
>
> | Pre-train Language  | CIDEr |
> | -------- | -------- |
> | EC  |   62.5 (1.4)  |
> | EC (shuffled)   |  59.2 (0.5)      |
> | paren-zipf    |  59.5 (0.2)     |
> | no pre-train  |  60.9 (0.5)|
>
>
> 4) **permuted EC corpora**
>
> Below is a comparison of pretraining on shuffled/unshuffled emergent corpus (we add the standard deviations for emergent language pretraining in Table 2). Though the gap between pre-training on emergent language and its shuffled version is not big, we observe that **such a gap is consistent at an individual level** (i.e. pre-training on one particular emergent corpus is always better than pre-training on its shuffled version), and the standard deviations at a population level are also small enough to test the effect. These are true also for the added image captioning experiments (see Response (3) above). We believe these findings do suggest that **emergent language structure is a consistent beneficial factor for downstream transfer (LM, image captioning)**. We also agree it would be an interesting future direction to study what shuffled EC could provide for transfer -- "token colocation statistics" could be a reasonable guess (see https://arxiv.org/pdf/2104.06644.pdf for a relevant discussion on MLM pretraining).
>
> |Pre-train Language  | LM (ro) | LM (he) |
> | -------- | -------- | -------- |
> | EC   |  198 (4)    |  375 (14)    |
> | EC (shuffled)    |  211 (1)        |     383 (4) |

---

> > ### Author Response · Authors · 2021-11-12
> > **Response to Reviewer G942 (continued)**
> >
> > 5) **Connections to related work about EC/NL multitasking**
> > Thank you for pointing out the related work about EC/NL multitask training, which will be addressed in the revised related work section. We agree this could be potentially more useful than just EC pretraining for NL tasks. However, our emergent language pretraining could be a clearer way to evaluate the emergent language, as mixing emergent and natural language during pre-training might add confounders to the transfer performance.
> >
> > 6) **Other ways of regularizing a language**
> >
> > We do not present these proposed regularization techniques in our paper as **we focus on the evaluation of emergent language** (see General Response (1)), and our pages are limited. In our experiments (e.g. 200 data points from Section 4) we do observe some degenerate cases where the EC game is not trained well and the transfer/translation performance is bad, and some regularization techniques might help. We believe our evaluation schemes provide a robust means to testing those regularization schemes.
> >
> >
> > 7) **standard deviations for EC pretrain/from scratch in Table 2**
> >
> > We are sorry these were missing in the draft, and here are the updated numbers with standard deviations, will which be incorporated into revision.
> >
> > |Pre-train Language  | LM (ro) | LM (he) |
> > | -------- | -------- | -------- |
> > | EC   |  198 (4)    |  375 (14)    |
> > | From scratch   |  265 (1)      |     446 (8) |
> >
> > 8) **Table 3 data points**
> >
> > As mentioned in the "Implementation" paragraph in Section 4, there are $5 \times 4 \times 10 = 200$ data points, because there are 5 game setups (sequence length and vocabulary size), 4 trials with different random seeds per setup, and 10 checkpoints per trial (training steps being $200, 400, \cdots, 2000$). We will include the scatterplot of 200 data points and how setups affect metrics and transfer performances in the appendix.
> >
> > 9) **Figure 3 understanding**
> >
> > Your understanding is correct.

---

> ### Author Response · Authors · 2021-11-18
> **Response to Reviewer G942: Revision Updated**
>
> Dear Reviewer G942,
>
> Thanks again for your constructive and encouraging comments. Today we have made substantial changes in the revision according to your review. In particular, we have the following changes according to **every rebuttal comment** you have raised (except the clarfication question you have about Figure 3):
>
>
> 1. **We have updated the paper to better address the comparison to model transfer**. We added the point that (i) transfer is also used in our work as a means to understanding and analyzing properties of emergent language beyond the game framework (Section 1), (ii) emergent language is a  representation with more general properties beyond parameters of particular EC models (Section 3.1), and (iii) more detailed discussion of significant practical advantages of corpus transfer over model transfer (Section 4.3), with added experiment results to confirm such an advantage even when corpus transfer uses the 1-layer GRU, and the clarficiation about the scalability of EC model architecture, thanks to your great suggestions. Finally, we summerize those points in Section 2 Related Work for a more comprehensive comparison to Li et al. (2020).
> 2. **Analysis of additional EC setups (vocab size, sequence length)**. We add in Appendix Section B.4 how vocabulary size and sequence length effect different metrics and downstream performances, with added results to support that larger vocabulary size/sequence length could lead to better downstream performance with less variance, which is best captured by our translation metric (over validation accuracy and topographic similarity). We appreciate your comment and believe the added results make our findings more robust and less particular.
> 3. **Additional image captioning results**. We added new ablation experiments in Appendix Section B.2 to confirm the non-triviality of image captioning results. Concretely, pre-training on emergent captions with shuffled tokens or ungrounded paren-zipf strings is worse than no pre-training, while emergent pre-training is better than no pre-training.
> 4. **Permuted emergent language**. We added in Section 4.3 that the gap between emergent language and permuted emergent language pre-training is consistent.
> 5. **Added discussion about EC/NL multitasking**. We added in Section 2 and Section 4.3 additional discussion about EC/NL multitasking for better EC or NLP task performance (instead of EC evaluation).
> 6. **Other ways of regularizing EC** We added related work you mentioned.
> 7. **Table 2 update**. We have updated Table 2 with standard deviations.
> 8. **Datapoints**. In Appendix B.3 we provide plots of 200 data points we used and their metric-downstream performance correlations.
>
>
> *We hope that the provided new experiments and additional explanations have addressed all your questions, and **would really appreciate it if you could raise your rating to 10.** Please let us know if you have further questions, or would like us to explore more into some new experiments.*
>
> Thank you for your time again!
>
> Best, Authors

---

> > ### Comment · Reviewer_G942 · 2021-11-22
> > **Thanks**
> >
> > Thanks to authors for the extensive response to my review and the updated experiments, all of which substantially alleviate my concerns I mentioned in the original review.
> >
> > I've read the other reviewers' comments, and while I think the additional experiments have strengthened the paper, I also recognize some of the limitations brought up by reviewer wvqW, in that the paper does not yet illustrate a setting where EC corpus pretraining would actually be used *in lieu of* the baselines explored. Specifically, Spanish outperforms all EC baselines. So if we want to do low-resource LM, in practice we should just use Spanish all the time, no?
> >
> > Nonetheless, I don't think this is a crucial weaknesses, and I don't think the paper as-is needs to have an immediate takeaway for people doing language modeling in production. If we had that standard, practically none of the work in emergent communication would ever get accepted, and this paper certainly makes a much greater step towards external usefulness than most in the literature. Thus, I'm still overall positive on the paper. It highlights additional ideas where EC might be more concretely useful in practice, e.g. perhaps combining transfer from other human languages (e.g. Spanish) with EC pretraining. It would be interesting to show such results, or similar results where we can actually overcome baseline performance, but I still think it's unreasonable to expect that much from a paper.
> >
> > To compare, Papadimitriou and Jurfasky (2020) have many experiments that show that pretraining on seemingly unrelated sources (music, code, parens) results in performance improvements, and more recently papers such as [Sinha et al., 2021](https://arxiv.org/abs/2104.06644) look at similar perturbations to the pretraining process. No one would ever honestly consider using such experiments in practice, yet these papers are still telling us something useful about the learning process (which may have later downstream implications). I view this paper as providing a similar story for the EC community.
> >
> > Overall I would like to keep my score at an 8 - I still vote for acceptance, given the author response and the other reviews and responses.

---

### Official Review · Reviewer_wvqW · 2021-11-03

**Correctness:** 2
**Technical Novelty And Significance:** 2
**Empirical Novelty And Significance:** 2
**Recommendation:** 3
**Confidence:** 4

**Details Of Ethics Concerns:**

I have no ethical concerns for this submission.

**Main Review:**

The paper's very clearly written, well-organized, and much of the discussion was grounded in a broad range of references to previous work.  My concerns with this paper are in regards the experimental design, the strength of the conclusions, and how the authors are choosing to interpret them.

Perhaps the best way to discuss this paper is through the authors' goals, to understand: unsolved: "(i) can emergent languages be used outside the game? (ii) if so, would these metrics predict the usefulness of emergent languages for downstream tasks? We tackle these two questions in Sections 3 and 4 respectively."

## can emergent languages be used outside the game?

To question (1), it is of course important to note that this has already been shown in Li et al. 2020 in the context of pre-training for MT.  There is plenty of opportunity to broaden the scope of their findings, but as it pertains to this question, it no longer needs to be asked and should be presented truthfully in this way.  This language persists throughout, and a reader not aware of Li's work would easily go through most of the paper believing that this paradigm is being attempted here for the first time.  For instance, even at the end of the paper "We present a new perspective for studying emergent communication by linking the corpora of emergent and natural languages in two ways."  One of these seems like an established perspective, and what sets this work apart is primarily the choice of tasks: language modeling and image captioning.

Returning to the question, can emergent languages be used outside the game for language modeling or image captioning?  First, it is also important to note that this EC setup is only one possibly game, and the properties of the EC are hugely shaped by the properties of the game.  So again, for accuracy, it's important to clearly define the scope of the work as pertaining to this one particular scenario.

For language modeling, the authors show that indeed the EC pre-training is useful for the downstream language modeling task, but the results are not very convincing that this would ever be useful in a practical sense.  Pre-training on EC data is rarely the best option, often outperformed by simple synthetic language data, and is never the best option as more data is observed.  And since pre-training on Spanish is still the most effective strategy even in low-resource languages, and Spanish is more easily acquirable than EC data, in what situation would one ever opt for using this strategy?  Back to the research question, yes, EC languages *can* be used outside the game, but it's not apparent that it would ever be best choice nor the easiest.

The case is similar for the captioning task, except the margins (even those between the best choice -- natural language -- and the worst choice baselines) is exceedingly small.  I think it throws into question whether MS-COCO was ever a good testbed for this research question, as (as the authors note) the captions are very regular and structured.  It seems like a reasonable attempt, but it's difficult to find a take-away.

When it comes to understanding the properties of the emergent languages, how these properties are translating into useful biases for downstream tasks, it is also left quite open.  The ablation experiments compare against random scenarios, and I personally wasn't able to gain much insight into where the performance comes from.  Personally I would have liked more qualitative analysis into the real correlations and differences of the EC and natural languages as it pertains to the downstream tasks, but even barring that I imagine there are also some purely quantitative metrics which could have been more targetted, and in doing so, be more informative.

## if so, would these metrics predict the usefulness of emergent languages for downstream tasks?

Another thread of this work is criticism of some existing EC metrics, like topographical similarity, which may be failing to correspond well to the types of structure and compositionality found in NLs.  The authors propose a new method of evaluating ECs by training (briefly) a translation model to translate to natural language captions from EC language generated from the same images.  The metric corresponds better to downstream task accuracy than topographic similarity.

The authors argue that "intuitively, a higher translation score means the emergent and natural sentences are closer in structure and semantics, similar to how French-English translation might be easier than Chinese-English".  But isn't the purpose of a such a linguistically-focused EC metric to evaluate how well EC reflects the types of structures found in natural (big-L) Language?  i.e., not any specific language.  It seems that in closely tracking the language used in the caption data, it is only natural that this relates to higher downstream task accuracy, but isn't that at the cost of low similarity to another natural language?  Criticisms of existing metrics are likely warranted, but I fail to see how this metric really improves upon it.  An EC learns a Chinese-like structure from a caption dataset, the translation model performs poorly at mapping it to English (vs a German-like EC source language) and now we have a plausible, NL-like EC with low scores under the metric.  I think ideally a good EC metric should score all human languages highly, or one should answer the question of which of two human languages contains more "human-like" language structure.  A metric that targets specific aspects of human languages cuts around this issue, but I think in this case the authors are burdened with providing some answer to this question.

So overall the story of the paper feels like a bit of circuitous reasoning: EC languages might be more helpful to downstream tasks than natural languages if they somehow capture a greater structural similarity to the target (presumably low-resource) language than available natural languages.  But the closer the language to existing natural language, the less useful it is, since it doesn't provide a unique advantage over simply using that natural language.  And in the space of all human languages, only the one closest in structure to the particular fine-tuning task achieves high marks.  So I'm curious how the authors reconcile this?


## Conclusion

Overall the work is tackling important problems with an interesting method, but in its current state I believe it's lacking clear insights into the problem, does not provide a practical usefulness, and doesn't dig sufficiently deep in understanding why models behave the way they do.  It also feels almost like two disjoint papers -- a paper could explore pre-training on ECs but with more focus given on understanding the properties of the ECs, and broadening it to more games to ensure lessons are more generally applicable.  Another paper could target metrics like topographical similarity, but in this case it is important to discuss the seemingly obvious failure cases of the proposed measure.

I think this is summed up well in the conclusion:
"Our methodology and results open up possibilities of establishing a synergy between the research of language emergence and natural language processing, where we could potentially better understand and improve emergent communication through the lens of natural language, and in turn improve upon low-resource natural language tasks with the help of emergent languages"

While the phrasing is again speculative/potential, I find it hard to view the paper experiments as strongly supporting either of these cases: I don't think we've learned much about the structure of ECs, even the limited examples used here in the paper, and I don't know of situations where EC would reasonably be used for improving low-resource monolingual NLP over NLs.

That's not to say the paper doesn't have a lot going for it, but each experiment felt like it only scratched the surface of the problem, and more thorough research feels necessary.



Typos / Grammar:
((Figure 3(a)))
"Our research in based on"



**Summary Of The Paper:**

This paper explores the use of pre-training models on corpora of emergent languages to improve performance on downstream tasks (in this case language modeling and image captioning).  The authors compare against using other types of pre-training corpora, like synthetic or natural languages.  While not always optimal, the authors demonstrate that there is clearly an advantage to using emergent languages in this way, if one is otherwise considering training from scratch.  They propose a new metric for evaluating whether languages have structure, doing so by training a translation model to map emergent language to natural language on the same data, and validate the measure by showing it corresponds closely to task accuracy.

**Summary Of The Review:**

The paper is well-structured and the experiments seem generally sound, but they're too brief and at too high of a level to provide real insight into EC language structure and how we should identify and measure it.  However, I do call into question really how sound the ablation experiments are, since they are contrasted with such trivial baselines that it again hinders the paper from providing novel insights.  There evaluation measure is I think the closest thing to a novel contribution, and the authors seem correct in pointing out the shortcomings of existing measures, but the proposed solution doesn't seem a suitable replacement (it seems like it violates some core assumptions of what the community would want from an EC metric, and is also language and task specific).  Overall it seems like good preliminary work, and I hope to see a deeper exploration using these same methods and motivations.

---

> ### Author Response · Authors · 2021-11-12
> **Response to Reviewer wvqW**
>
>
> Thank you for the critical comments! We provide detailed responses to your questions below.
>
> ### Part I: can emergent languages be used outside the game?
>
> 1) **Novelty and claims in the context of Li et al. (2020)**
>
> We thank the reviewer for raising this important issue.
>
> - As argued in detail in **General Response (2)**, our corpus transfer approach is fundamentally different from the model transfer approach in Li et al. (2020).
>     - First, even though both works connect EC games to downstream NLP tasks, **Li et al. (2020) mainly aim to improve NLP task performance, whereas we are the first work to envision transfer as a means for emergent language evaluation and analysis**, which is a paradigm shift compared to previous metrics that operate within the game framework and provide limited insights toward natural language properties.
>     - Second, **our research objective is not focused on the EC models, but the emergent language produced by the EC game**, which could be learned and used by new agents with properties beyond the parameters of a particular speaker model. We are the first work to study the emergent language and its properties by transferring to NL tasks, and that is why we ask "can emergent languages be used outside the game?" (not addressed by Li et al. (2020)) instead of "can emergent communication (models) be used outside the game?" (addressed by Li et al. (2020)). We will better stress the differences in revision.
>     - Last but not least, corpus transfer also bears significant practical advantages over model transfer. Please refer to **General Response (2)** for more details, where we present new evidence that corpus transfer (even just with one-layer GRU instead of the six-layer Transformer used in the paper) outperforms model transfer.
>
> - We have strived to give due credit to Li et al. (2020) by citing this reference throughout the paper 7 times across four out of six sections, discussing its inspiration to our work (intro, related work), practical limitation (corpus transfer section) and relevant findings (translation section). In light of ours differences, we have intentionally quantified our methodology, contributions, and claims. For example, we highlight our approach being "corpus transfer" as early as the title and explicitly contrast it with prior "model transfer" throughout the paper. Also, we have consistently expressed emergent language (instead of emergent communication or emergent communication models) as the concrete research subject we wish to study, evaluate or leverage for NL tasks.
> - We do, however, realize that our comparison to Li et al. (2020) in the current draft mainly leans toward practical performance, which might mislead readers in terms of our conceptual novelty and contributions. We plan to further explain in related work how our motivation and scientific focus are fundamentally different from  Li et al. (2020), as well as add discussion and new evidence to the practical advantages of corpus transfer over model transfer.
>
> 2) **EC setup is particular**
> - Our *basic* setup (game input, training signal, optimization technique) of EC is particular (and follows established prior work like Li et al. (2020) and Lee et al. (2018)), but this is true for most, if not all, other EC papers. Meanwhile, we do vary *other properties of the EC game* (e.g. vocabulary size, sequence length, training steps) in Section 4.  We will include more details about those different EC trials and how different setups affect metrics and downstream performances in the appendix (please see Response to Reviewer gtt4 (2) for a concrete example).
> - While previous EC papers tend to investigate different game setups and how they affect certain metrics, we note that such a paradigm cannot provide robust and valuable insights if the metrics are flawed. Thus, as argued in General Response (1), our work focuses on the evaluation (but not improvement) of emergent language. Our progress in this direction would enable future works to more readily investigate and improve EC setups.
>
> (to be continued)

---

> > ### Author Response · Authors · 2021-11-12
> > **Response to Reviewer wvqW (continued)**
> >
> >
> > 3) **Pre-training on EC data is rarely the best option**
> >
> > - The direction of leveraging EC for NLP tasks is very recent, and **we are the first work to show any evidence that under the same corpus size, an emergent language corpus could provide greater transfer benefit compared to a natural language**. Such a finding, despite being preliminary, is very surprising in the context of prior work (Li et al. (2020) do not compare to NL pretraining, while in Papadimitriou & Jurafsky (2020) no artificial languages (music, code, regular languages) could transfer better than natural languages), and should be interesting and relevant for both EC and NLP communities.
> > - As mentioned in the "Limitations and Future Directions" part of our paper, **our work could serve as pioneering work to inspire future work that pushes EC to be more useful for NLP and ML**. Some potential directions where pre-training on EC could be much more useful include
> >     - incorporate existing techniques that improve the emergent language to improve emergent language pre-training;
> >     - leverage EC pretraining for embodied tasks (e.g. Alfred) where **NL annotation is hard to collect and scale**;
> >     - mix EC and NL pretraining for better performance than just NL pretraining (e.g. https://aclanthology.org/2020.acl-main.685.pdf pointed out by Reviewer gtt4);
> >
> >
> > 4) **The margin of image captioning results**
> > Our focus for the image captioning experiment is toward the **low-resource setup**, i.e. when natural language annotations are limited (5k or 50k captions), EC-pretraining could significantly improve performance (e.g. increasing BLEU4 score by around 1). Note that increasing BLEU4 score by 1 is considered highly non-trivial in the image captioning community, where top models often only differ by less than 0.5 in BLEU4. We believe **such a low-resource setup better simulates the intended applications for emergent language pre-training and serves as a starting point to inspire future efforts toward tasks where natural language annotations are truly scarce or hard to collect** (e.g. low-resource languages, embodied tasks that are hard to annotate by humans).
> >
> >
> > 5) **more qualitative analysis into the real correlations and differences of the EC and natural languages**
> >
> > We are not sure what "real correlations and differences of the EC and natural languages" mean exactly. We will add some examples of images and their emergent and natural captions in the appendix for qualitative display. However, we note that our paper intentionally aims to move away from previous manual-check paradigms in EC papers and tries to establish a more standardized, scalable, and robust quantitive metric to enable progress in this field.
> >
> > 6) **some purely quantitative metrics which could have been more targetted**
> >
> > Please see Response (9) below, in the context of the translation metric.
> >
> > 7) **The ablation experiments compare against random scenarios, and I personally wasn't able to gain much insight into where the performance comes from**
> >
> > We are not sure what "random scenarios" mean exactly. Like ablation studies in any other paper, our ablation experiments in Section 3 aim to confirm basic components (e.g. perceptual input and communication training in EC game, sentence structure in EC corpus, how to transfer) matter for the final NL task performance, and from the numbers we can also understand better how these components matter differently, e.g. Table 2 indicates communication training is more important than perceptual input in the EC game for downstream performance. Section 4 provides more experiments with different game setups (vocabulary size, sequence length, training steps), and we will add findings of how setups affect metrics and transfer in the appendix (please see Response to Reviewer gtt4 (2) for a concrete example).
> >
> > (to be continued)

---

> > > ### Author Response · Authors · 2021-11-12
> > > **Response to Reviewer wvqW (continued)**
> > >
> > >
> > > ### Part II: if so, would these metrics predict the usefulness of emergent languages for downstream tasks?
> > >
> > > 8) **Translation metric ties to a particular natural language**
> > >
> > > We agree our translation metric only approximates how close an emergent language is to natural (big L) Languages, but this is true for all artificial metrics (e.g. ELBO approximates likelihood). In our case, it is important to note that the approximation is twofold:
> > > - **The performance of a neural translation model only approximates (upper bounds) the distance between two languages**. Imagine using some oracle translation model (or highly professional human translators), it is likely that Chinese/French to English translation yields similarly high scores, and all natural languages are very close to each other. Then tying to one particular NL is less concerning.
> > > - **One particular NL (English in our case) only approximates properties of all natural Languages**. But given current emergent languages are still very different from ANY natural language,  arguably moving toward one NL would provide coarse gradients toward all natural Languages. In fact, Table 3 confirms being closer to English is consistently helpful for modeling two other very different natural languages, Romanian and Hebrew. We are happy to include more downstream languages if needed.
> > >
> > > 9) **A metric that targets specific aspects of human languages**
> > >
> > > This is definitely an interesting and important future direction, and it is the goal of our paper to inspire such research questions previously unraised. We note that **transfer to different downstream NL tasks** is one way to focus on different NL properties. For example, our LM experiments focus more on structural properties of the source language, while image captioning focuses more on semantic properties (more evidence about this in Response to Reviewer G942 (3)). Following the previous translation metric discussion, it might also be possible to **constrain the translation model in different aspects** to make translation results reflect targetted properties. For example, lexical-level translation might evaluate more about the alignment of emergent and natural language structures. We are excited about the potential enabled by our work to further explore such a direction.
> > >
> > > 10) **the closer the (emergent) language to existing natural language, the less useful it is, since it doesn't provide a unique advantage over simply using that natural language**
> > >
> > > We respectfully disagree. The ultimate goal of the EC research community has always been to push emergent languages to be as close to natural language as possible. This would not only answer fundamental scientific questions about how natural languages emerge and evolve, but also entail a huge practical impact in NLP and AI - **an artificial model that can utter and use natural language could be much more useful than static natural language corpora**. Just for one example, think about embodied tasks (or even just image captioning) where the human annotation is time-consuming and hard to scale. A "perfect" EC speaker (that utters natural language) would essentially enable unlimited and automatic natural language annotations for such tasks.
> > >
> > > 11) **only scratched the surface of the problem**
> > >
> > > To summarize, there are indeed a lot of very important future directions (better emergent language, better emergent language pretraining, more targetted metrics, etc.), but **our work is about the paradigm shift that enables these questions to be raised properly in the first place**. We believe our work "seems like good preliminary work" exactly because it opens up many possibilities of "a deeper exploration using these same methods and motivations".

---

> > > > ### Comment · Reviewer_wvqW · 2021-11-22
> > > > **Reviewer Response**
> > > >
> > > > Thank you, authors, for the thorough response.  There's a lot to unpack, and I'm sorry it has taken so long to get around writing a reply.  Because of the timetable and the breadth of the reply, I think it will likely be better to make a couple concise-ish arguments vs. a more point-by-point rebuttal.
> > > >
> > > > 1.  Regarding corpus transfer vs. model transfer, since this is intended to be a salient point in the paper, I found it surprising how much of the experimental sections are geared towards comparisons to NLs and baselines that didn't involve a model transfer component, and how thin the comparison to model transfer is.  Because of this, I was getting a bit sidetracked by my own interpretation of Section 3, and I think I agree overall with the rebuttal and the earlier posted comment by reviewer gtt4 that there is then maybe then no need to bring Li et al. further into the discussion.
> > > >
> > > > As for the comparison between corpus transfer and model transfer, I think the experiments reported in the response seem reasonable and help in this regard.  I think it should still be noted that these are not really valid comparisons (closer on the GRU vs GRU, but not much of win their either), but the fact that they are not valid is really the strength of the method -- good performance comes from leveraging EC into models people have difficulty doing EC with (so far).
> > > >
> > > > There's still a real problem in my mind though for when I would use such a method.  I know that there have been papers along similar lines, making appeals to future scenarios where there's a low resource language (but we have captions?) and where the EC fills these gaps, and I have read the scenarios in the rebuttal.  But in a lot of comparisons even quite trivial baselines are performing quite competitively with the proposed method -- that should really give some cause for alarm.  I like the intuitions of this paper, but these more practical appeals for EC really require what to me is an unrealistic leap of faith.  The paper would be so much stronger in my eyes if it had enough conviction in these types of scenarios to tackle one in the paper.  Or barring that, to try to find another potential motivation for downstream performance from EC models -- I wonder if it makes more sense in a game world than in real world where I think we could feasibly acquire samples of any language we would ever be able to do such a task for.
> > > >
> > > >
> > > > 2.  My bigger gripe is with the evaluation proposal.  It just misses the mark to me.  That is not to say that existing EC metrics don't have their flaws - they do - but I also felt the community viewed these as stop-gap measures to eventually devise more effective measures to examine EC languages in detail.  This metric takes the opposite approach, and now tucks many of details further away within the weights and operations of a translation model.  It no doubt tracks closely to downstream performance, as it's rewarding ECs that are close to the downstream task language, but if the authors want to really propose it as a useful metric for studying EC, I think more experiments and more considerations are necessary.  There's a lot of properties of language that are often talked about in EC (say compositionality/hierchical structure) that are difficult to translate, and difficult to utilize in many downstream tasks.  Wouldn't there also be biases related to what type of translation model is used and what sorts of syntax and morphology would be reflected in the measure?  Or even the size of the model?  And what about imbalances between source and target in terms of under/overspecification.  And what types of information are actually useful to the downstream tasks now (i.e., what are these language capturing)?  I don't think we have any idea.  That is not to say the idea doesn't have merit, but because it's bringing in more complexity, I think it brings in a lot of confounds that need to be eliminated, or at least explored and discussed.
> > > >
> > > > So I think there are many factors that come into play when proposing a metric of this sort that relies on both an extra model and an extra language, and I don't think any of them have been explored much at all.  To call it a paradigm shift and push these questions to future work is I think not the appropriate move.
> > > >
> > > > In many ways this paper feels well on its way to being two papers, but in doing so it comes up a bit short on each of its main contributions (+ some points if one can see a practical importance of EC pre-training on downstream tasks).  I think it's no surprising then to have scores split so drastically - if both of these ideas hit, there's really a lot here, and it's smart and well-written.  I will still follow the discussion and consider changing my score.  I'm also curious if any of the other reviewers share my general skepticism for the translation-based evaluation measure.

---

> ### Author Response · Authors · 2021-11-18
> **Response to Reviewer wvqW: Revision Updated**
>
> Dear Reviewer wvqW,
>
> Thanks again for your detailed comments. We have made substantial changes in the revision according to your review. In particular, we have the following changes according to your comments.
>
>
> 1. **We have updated the paper to better address our fundamental differences with Li et al. (2020)**. We added the point that (i) transfer is also used in our work as a means to understanding and analyzing properties of emergent language beyond the game framework (Section 1), (ii) emergent language is a  representation with more general properties beyond parameters of particular EC models (Section 3.1), and (iii) more detailed discussion of significant practical advantages of corpus transfer over model transfer (Section 4.3), with added experiment results to confirm such an advantage even when corpus transfer uses the 1-layer GRU. Finally, we summerize those points in Section 2 Related Work for a more comprehensive comparison to Li et al. (2020).
> 2. **Analysis of additional EC setups (vocab size, sequence length) (Appendix B.4)**. We add in Appendix Section B.4 how vocabulary size and sequence length effect different metrics and downstream performances, with added results to support that larger vocabulary size/sequence length could lead to better downstream performance with less variance, which is best captured by our translation metric (over validation accuracy and topographic similarity). We appreciate your comment and believe the added results make our findings more robust and less particular.
> 3. **More discussion on better EC pretraining**. According to our rebuttal, we added more concrete discussion about how to improve EC pre-training (end of Section 4.1), and point to this along with the end of Section 4.2 in Section 6 Conclusion.
> 4. **Additional image captioning results (Appendix B.2)**. We added new ablation experiments in Appendix Section B.2 to confirm the non-triviality of image captioning results. Concretely, pre-training on emergent captions with shuffled tokens or ungrounded paren-zipf strings is worse than no pre-training, while emergent pre-training is better than no pre-training.
> 5. **Qualitative examples and analysis (Appendix B.5)**. We added in Appendix B.5 some examples of images and their natural and emergent captions, along with some simple analysis.
> 6. **Updated conclusion**. In light of the rebuttal, we updated Section 6 Conclusion to better reflect our position and how our work could inspire important and exciting future directions (better EC, better EC pre-training, more fine-grained metrics).
>
> **We wish that our response has addressed your concerns, and turns your assessment to the positive side.** *If you have any questions, please feel free to let us know during the rebuttal window.*
>
> Thank you again!
>
> Best, Authors

---

> ### Author Response · Authors · 2021-11-23
> **Second Response to Reviewer wvqW**
>
> Thank you for replying and acknowledging that we have addressed the comparison to Li et al. (2020), one of the main concerns in your initial review.  Here are further responses to your remaining concerns.
>
> ### I. Usefulness of EC pre-training
> We appreciate the continuing discussions on the external usefulness of emergent communication. This fundamental issue pertains not merely to our work, but to the research field of EC at large that bets on its promise to eventually enable usefulness and progress in AI. For many years, such a promise has been elusive, still an entire community sees the bet as worth making and pushes for progress.
>
> Now, our work takes a significant step toward this promise with the novel methodology of corpus transfer. We choose to first establish the method in common and standardized tasks like language modeling and image captioning, which have been studied by numerous reseachers with rich NL resources, and are indeed the most challenging tasks for any methods to improve upon. However, by showing the potential of EC pre-training with a low-resource setup of these most studied tasks, we aim to appeal to a more general audience and lay an indispensible foundation for studying more specialized tasks that we exemplify below.
>
> - **Tasks where data is significantly different from NL, and/or human annotation is extremely hard**, e.g. protein or DNA tasks where any human annotation requires highly professional knowledge and lab work. Pre-training on an emergent language that *grounds* on these data sources in an unsupervised way could be more beneficial than pre-training on *ungrounded* natural language with very different structural and semantic properties.
> - **Tasks where machines could already outperform humans**, e.g. StarCraft. Collecting human annotations about how to coordinate and communicate in this game is not just tedious and laborious (and unrealistic given the game is played in real-time), but also possibly inferior to emergent communication annotations from better AI agents, when transfering to a similar yet novel game.
> - **Tasks where machines need to communicate**, e.g. the task in Lazaridou et al. (2020) where a machine speaker needs to communicate with a human listener in an image reference game. They show a mix of NL and EC training could be more beneficial than NL training alone: while training the machine speaker on (NL) image captioning data gains general semantic and linsuistic knowledge, training in an EC game with a machine listener provides complementary communication skills like pragmatics that are otherwise hard to obtain from NL annotations --- collecting human-human communications might be less useful due to the distributional shift between human and machine speakers, and collecting machine-human communications is tortuous for humans when the machine speaker is not yet good enough, and is less reusable as each machine-human coordination could be different.
>
> No matter which potential direction to go, we hope to point out the value of our study that starts with basic ML/NLP tasks --- just like how Li et al. (2020) study translation and Papadimitriou et al. (2020) study LM --- and paves the way for future tasks.
>
> ### II. The translation metric
> We and you both agree that previous flawed EC metrics measure **[A] specific properties of "something" non-NL** (e.g. "functionality" within a particular game, or over-approximated "compositionality"), while the ultimate ideal EC metric would measure **[B] specific properties that link to NL**. However, we respectually disagree that toward "eventually devise more effective measures to examine EC languages in detail", we are taking "the opposite approach". Rather, our contribution is to point out **the current bottleneck is not the type or specificality of properties, but the linkage to NL** --- when properties do not link to NL, **[C] even further investigating properties within the EC game** would not provide any reliable insights about NL. Thus, we introduce the first metric on **[D] general properties that link to NL**, which already better predicts transferability to NL with insights not enabled by previous metrics (e.g. Figure 3).
>
> During the rebuttal, we have "explored and discussed" possible extensions from our translation metric. For example, Appendix B.4 controls the confound of vocabulary size and sequence length and shows the translation metric still provides insights that better link to NL transfer. And as argued in the previous response (9), your concern about using different translation models is exactly an opportunity to study more specific NL properties. In summary, our translation metric _[D]_ is not at the finish line _[B]_ yet, but is about the shift from the previous flawed direction _[A]_ that only leads to an dead end _[C]_, to a more promising path that provide insights along the way.
>
> (to be continued)

---

> > ### Author Response · Authors · 2021-11-23
> > **Second Response to Reviewer wvqW (continued)**
> >
> > ### III. The two parts of the paper
> > As argued in (I, II) above and throughout the rebuttal, further developing either part of the paper is possible and promising. However, we hope to point out that both parts in our work are logically connected and indispensable to argue for our main message: **a shift from flawed in-game EC metrics to those actually linked to NL properties**.
> >
> > To argue previous metrics are flawed, we need to establish an justifiable end to building metrics, i.e. predict the usefulness for NL tasks (through corpus transfer). To present an initial idea for metrics that link to NL, we propose the translation metric and show it indeed better predicts transerability and provides concrete insights. **Both parts are brand new rather than incremental**, thus require extensive establishment, and **we did our best to convey this main message with a logically complete set of discussions and empirical results that suit a page limit of 9**. Within such a modest page limit, any branching into either part more would inevitably cut the other part and jeopardize our main message, and we have strived to extend both parts with new discussions and experiments during the rebuttal period and revised them into the paper and appendix.
> >
> >
> > *Please don’t hesitate to let us know if there are any additional clarifications that we can offer, as we would love to convince you of the merits of the paper.*
> >
> > *Thank you for your time again!*
> >
> > Best, Authors
> >
> > ### References
> > - Li et al. Emergent Communication Pretraining for Few-Shot Machine Translation. COLING 2020.
> > - Papadimitriou et al. Learning Music Helps You Read: Using Transfer to Study Linguistic Structure in Language Models. EMNLP 2020.
> > - Lazaridou et al. Multi-agent Communication meets Natural Language: Synergies between Functional and Structural Language Learning. ACL 2020.

---

> > > ### Author Response · Authors · 2021-11-26
> > > **Sincerely Look Forward to Further Discussions!**
> > >
> > > Dear Reviewer wvqW,
> > >
> > > Happy Thanksgiving! Hope you enjoyed a wonderful holiday.
> > >
> > > As the deadline for discussion is approaching, we want to make sure if our second response addresses your remaining concerns. We would love to further engage in the limited remaining time if needed, to the best of our ability.
> > >
> > > Thanks again for your time!
> > >
> > > *Best, Authors*

---

### Author Response · Authors · 2021-11-12
**General Response**

We thank all reviewers for their comments and suggestions! Here we reiterate the position of our work, and our fundamental differences to Li et al. (2020).

1) **Scope of our work**

Though at the interaction of EC and NLP, our work is mainly rooted in the EC community and calls for **a paradigm shift for emergent language evaluation**, as we see evaluation as the bottleneck of current language emergence research. We believe **such a goal is a complement to, yet also a prerequisite of, properly pursuing several other important directions in EC research**:

* **Improve/regularize emergent communication/language (Reviewers wvqW, G942, gtt4)**. While previous EC papers tend to investigate different game setups and how they affect certain metrics to draw high-level insights, we believe that **such a paradigm cannot provide robust and valuable insights if the metrics themselves are flawed**. Thus our work focuses on the evaluation (but not improvement) of emergent language. Progress in this direction would enable future works to more readily investigate and improve EC setups. That being said, **we will also add analysis into how our game setups (vocabulary size, sequence length, training step) affect metrics and downstream performances in the appendix. For a concrete example, see Response to Reviewer gtt4 (2).**

* **More fine-grained evaluations for emergent language based on different properties of NL**. By providing insights into flaws of previous metrics and introducing the idea of evaluating emergent language by linking to natural languages, our work serves as a starting point for more fine-grained evaluations and analysis for emergent language that target different NL properties. See Response to Reviewer wvqW (9) for more discussion.

* **Improving NLP performance with EC**. This direction of leveraging EC for NLP tasks is very recent, and **we are the first work to show any evidence that under the same corpus size, an emergent language corpus could provide greater transfer benefit compared to a natural language**. Such a finding is very surprising in the context of prior work (Li et al. (2020) do not compare to NL pretraining, while in Papadimitriou & Jurafsky (2020) no artificial languages (music, code, regular languages) could transfer better than natural languages), and should be interesting and relevant for both EC and NLP communities. **We believe our methodology and findings could inspire future efforts that push EC to be more useful for tasks with limited natural language resources (see more in Response to Reviewer wvqW (3))**.

2) **Corpus transfer vs. model transfer**


Our main difference from Li et al. (2020) is that Li et al. (2020) directly transfer *emergent communication models* for downstream tasks (model transfer), while we use a corpus of *emergent language* outside the game. Such a difference is significant for the following reasons.

* **Motivation**. Even though both works connect EC games to downstream NLP tasks, **Li et al. (2020) mainly aim to improve NLP task performance, whereas we are the first work to envision transfer as a means for emergent language evaluation and analysis**, which is a paradigm shift compared to previous metrics that operate within the game framework and provide limited insights toward natural language properties.

* **Scientific focus**. Our core research goal is to evaluate **languages** developed by communication games. A language could be learned, spoken, and used by new agents (e.g. the Transformer in our experiments), and should not be tied to a speaker with some particular architecture and parameters. This is analogous to how linguists might study English literature and books rather than one particular person that speaks English.

* **Practical advantages**. Because new models could be learned and engaged with the emergent language, our corpus transfer approach has several practical advantages over the model transfer approach.

    * In many downstream tasks (e.g. image captioning with detection features as input), the speaker and listener architectures might not be directly applicable, while corpus transfer is still easy to deploy and benefit from.
    * In other downstream tasks (e.g. language modeling) where the speaker and listener architectures could be used, corpus transfer allows greater transfer performance by using more powerful architectures, which is shown in Table 3.
        * In addition, we note that, unlike most NLP tasks, architectures for EC agents cannot be arbitrarily big/expressive (Response to Reviewer G942 (1)), which limits model transfer to leverage more powerful models.
        * Even when corpus transfer does not leverage more powerful architectures, new experiments show it might still outperform model transfer (Response to Reviewers G942 (1), gtt4 (3)).

In revision, we will address our differences more comprehensively in terms of motivation, scientific focus, and additional evidence of empirical advantages.

---

### Author Response · Authors · 2021-11-18
**General Response: Revision Updated**

Dear Reviewers:

Thank you all again. We would like to restate our key novelty and contribution: **a paradigm shift for how we evaluate and analyze emergent language, through transfer and translation to natural language**, which is drastically different from prior work like Li et al. (2020) and could enable and inspire several exciting future directions (e.g., better EC, better EC pre-training, more fine-grained metrics for NL properties).

In the previous rebuttal response, we have provided detailed comments to each reviewer's questions and concerns with new experiment results and clarficiations. **Now, we have revised our paper with substantial changes to address all of your questions.**

- (Reviewer wvqW, G942, gtt4) **More comprehensive comparison with related work, especially Li et al. (2020)**. We move Related Work up to Section 2 to better clarify our differences with related work earlier in the paper, and add missing work pointed out by reviewers. Specifically, we summerize points we made in General Response for a more comprehensive comparison to Li et al. (2020). **We believe such updates significantly better reflect our novelties and contributions**.
    - transfer is also used in our work as a means to understanding and analyzing properties of emergent language beyond the game framework (Section 1);
    - emergent language is a  representation with more general properties beyond parameters of particular EC models (Section 3.1);
    - (Reviewer G942, gtt4) more detailed discussion of significant practical advantages of corpus transfer over model transfer (Section 4.3), with added experiment results to confirm such an advantage even when corpus transfer uses the 1-layer GRU.
- (Reviewer wvqW) **More discussion on better EC pretraining**. According to our rebuttal, we added more concrete discussion about how to improve EC pre-training (end of Section 4.1), and point to this along with the end of Section 4.2 in Section 6 Conclusion.
- (Reviewer G942, gtt4) **Updated Tale 2**. We have updated Table 2 with standard deviations, and used dotted red lines for training from scratch with text explanations.
- (Reviewer wvqW) **Updated conclusion**. In light of the rebuttal, we updated Section 6 Conclusion to better reflect our position and how our work could inspire important and exciting future directions (better EC, better EC pre-training, more fine-grained metrics).
- (Reviewer G942) **Additional image captioning results (Appendix B.2)**. We added new ablation experiments in Appendix Section B.2 to confirm the non-triviality of image captioning results. Concretely, pre-training on emergent captions with shuffled tokens or ungrounded paren-zipf strings is worse than no pre-training, while emergent pre-training is better than no pre-training.
- (Reviewer G942) **Datapoints (Appendix B.3)**. In Appendix B.3 we provide plots of 200 data points we used and their metric-downstream performance correlations.
- (Reviewer wvqW, G942, QkKj, gtt4) **Analysis of additional EC setups (vocab size, sequence length) (Appendix B.4)**. We add in Appendix Section B.4 how vocabulary size and sequence length effect different metrics and downstream performances, with added results to support that larger vocabulary size/sequence length could lead to better downstream performance with less variance, which is best captured by our translation metric (over validation accuracy and topographic similarity). **We believe the added results make our findings more robust and less particular.**
- (Reviewer wvqW) **Qualitative examples and analysis (Appendix B.5)**. We added in Appendix B.5 some examples of images and their natural and emergent captions, along with some simple analysis.



Please don’t hesitate to let us know of any additional comments on the manuscript or the changes!

Best, Authors

---

### Author Response · Authors · 2021-11-19
**Thanks for all your comments and look forward to post-rebuttal feedbacks!**

Dear AC and all reviewers:

Thanks again for all of your constructive suggestions, which have helped us improve the quality and clarity of the paper!

Since the discussion phase has started for over one week, we have not heard any post-rebuttal response yet.

Please don’t hesitate to let us know if there are any additional clarifications or experiments that we can offer, as we would love to convince you of the merits of the paper. We appreciate your suggestions. Thanks!

---

### Decision · Program_Chairs · 2022-01-20

**Decision:**

Accept (Spotlight)

**Comment:**

This paper explores ways in which *emergent communication* (EC) methods from representation learning can be evaluated extrinsically, by hooking them into downstream NLP tasks. Reviewers agree that the paper is thorough, and finds encouraging results.

This paper is borderline, and difficult to evaluate, even after very substantial discussion (some of it private). From my reading of the reviews and pieces of the paper, I'm very sympathetic to wvqW's concern that none of the present-day applications under study seem likely to benefit from this kind of emergent communication pretraining: *Natural* language pretraining, even transferring across natural languages, is for too strong a baseline, and it's not even conceptually clear how one could substantially outperform that baseline. I'm very concerned that the results in this paper will be—misleadingly—cited as proof that EC research is already contributing to downstream progress in NLP.

However, the narrow claims in the paper itself seem to be sound, and two confident reviewers whom I trust argue strongly that the ideas results here are surprising and novel, and that the paper could be the starting point for productive discussion and future work in this area. I'm recommending spotlight presentation in the hope that the paper will provoke a nuanced discussion in that setting.